# Nitrogen and phosphorus losses by surface runoff and soil microbial communities in a paddy field with different irrigation and fertilization managements

**Limin Wang[1,2], Dongfeng Huang📷[1,2]***

**1** Soil and Fertilizer Institute, Fujian Academy of Agricultural Sciences, Fuzhou, P. R. China, **2** Fujian Key Laboratory of Agro—products Quality & Safety, Fujian Academy of Agricultural Sciences, Fuzhou, P. R. China

* hdf_2169395@126.com

**Data Availability Statement:** All microbial files are available from the Biotechnology Information (NCBI) Sequence Read Archive (SRA) database with accession number SRP293735.

## Abstract

Rice cultivation usually involves high water and fertilizer application rates leading to the non-point pollution of surface waters with phosphorus (P) and nitrogen (N). Here, a 10-year field experiment was conducted to investigate N and P losses and their impact factors under different irrigation and fertilization regimes. Results indicated that T2 (Chemical fertilizer of 240 kg N ha$^{-1}$, 52 kg P ha$^{-1}$, and 198 kg K ha$^{-1}$ combined with shallow intermittent irrigation) decreased N loss by 48.9% compared with T1 (Chemical fertilizer of 273 kg N ha$^{-1}$, 59 kg P ha$^{-1}$, and 112 kg K ha$^{-1}$ combined with traditional flooding irrigation). The loss ratio (total N loss loading/amount of applied N) of N was 9.24–15.90%, whereas that of P was 1.13–1.31% in all treatments. Nitrate N ($NO_3^-$−N) loss was the major proportion accounting for 88.30–90.65% of dissolved inorganic N loss through surface runoff. Moreover, the N runoff loss was mainly due to high fertilizer input, soil $NO_3^-$−N, and ammonium N ($NH_4^+$−N) contents. In addition, the N loss was accelerated by *Bacteroidetes*, *Proteobacteria*, *Planotomycetes*, *Nitrospirae*, *Firmicutes* bacteria and *Ascomycota* fungi, but decreased by *Chytridiomycota* fungi whose contribution to the N transformation process. Furthermore, T2 increased agronomic N use efficiency (AEN) and rice yield by 32.81% and 7.36%, respectively, in comparison with T1. These findings demonstrated that T2 might be an effective approach to ameliorate soil chemical properties, regulate microbial community structure, increase AEN and consequently reduce N losses as well as maintaining rice yields in the present study.

## Introduction

Rice (*Oryza sativa* L.) is one of the main staple crops and feeds over 65% of the world's population with 11% of cultivated land [1,2]. Because the population is steadily increasing, rice production must increase by 1% annually [3]. High rice yields depended on higher inputs of

**Funding:** This work was supported by the Special Fund of Fundamental Scientific Research at Nonprofit Research Institutions in Fujian (2018R1022-4), Innovation Team in Fujian Academy of Agricultural Sciences (STIT2017-2-10), Fuzhou Science and Technology Support Program (2018-G-65), the Youth Talent Program of Fujian Academy of Agricultural Sciences (YC2019006), the Open Research Fund of Fujian Key Laboratory of Agro - products Quality & Safety, China (APQSKF201902), and the Natural Science Foundation of Fujian Province, China (2020J011358).

**Competing interests:** The authors have declared that no competing interests exist.

nitrogen (N) and phosphorus (P) fertilizers, however, which inevitably increased the risk of potential eutrophication in the surrounding water bodies through surface runoff from paddy soils [3,4]. Eutrophication is the excessive growth of algae in response to N and P additions and consequently leads to a heavy mortality of other aquatic plants and animals resulting from the decomposition of algae [4]. To date, water—quality deterioration as a consequence of eutrophication was observed in many regions such as Europe, America and China [4–6]. In addition, a main N and P loss pathway is the direct loss of manure, fertilizer and/or soil to surface water by runoff [7]. Moreover, surface runoff is determined primarily by high irrigation and precipitation events [8]. Minimizing N and P concentrations in runoff is therefore important to protect receiving waters from eutrophication. A widely used method to achieve this is to optimize water and fertilizer management. For example, water-saving irrigation techniques could maintain rice yields despite 50% of the irrigation volume, compared to traditional irrigation [9]. Reduction of chemical fertilizer input is also a potential solution to lower nutrient export fluxes [10]. Therefore, nutrient runoff losses could be reduced by optimizing fertilizer and water management practices during the rice growing seasons.

The wet—dry cycles of water saving irrigation combined with optimizing fertilization also changed soil properties, N and P transformation. These changes directly resulted in different characteristics of N and P use efficiency and loss from paddy fields [11]. It has been reported that soil moisture and temperature were important factors influencing seasonal variations in losses of available N and P in simulated freeze-thaw conditions [12]. In addition, optimal applications of water and fertilizers affected soil microbial communities, consequently leading to variations in N and P losses by surface runoff in field conditions [13,14]. Related studies have suggested that arbuscular mycorrhizal fungi (AMF) can not only scavenge P resources by improving P uptake of rices, but also reduce N losses from paddy soils through denitrification [15,16]. The combination of inoculation with AMF and 80% of the local norm of fertilization reduced N runoff by 27.2% [17]. Additionally, ammonia-oxidizing bacteria (AOB) played an important role in the ammonia oxidation which was crucial for N and P runoff losses [18]. These N cycling processes were closely linked to N and P losses. Hence, understanding the response of microbial communities to fertilization and irrigation is important to select the optimum water and fertilizer management to minimize nutrient inputs in paddy soils.

Soil microbial community composition and diversity were reportedly altered over a wide range of soil factors associated with water and fertilizer managements [14,18]. The present studies have mostly focused on the impacts of either irrigation management or fertilizer application alone on the microbial communities [14,19], but few studies have evaluated microbial community structure in response to the combination of water and fertilizer management, particularly in subtropical paddy soils. However, different irrigation and fertilization regimes tended to shape distinct microbial communities [14,19]. In addition, N and P runoff losses varied temporally, and little information about nutrient runoff losses from paddy fields was available in this region. The subtropical paddy field is one of the major rice production bases of South China. Importantly, the rice growing season in this area extends from May to September each year which corresponds with the main rainy and hydrologically active period of the year. The surrounding water bodies were vulnerable to pollution from N and P in paddy fields. To date, N and P runoff losses and their influencing factors while maintaining or enhancing rice yields in the paddy fields in southeastern China are currently unclear under different irrigation and fertilization regimes. Thus, we hypothesized that different irrigation and fertilization practices could alter soil chemical properties and microbial community structure, which would subsequently affect N and P runoff losses. To test the hypothesis, a 10-year plot experiment was conducted to estimate N and P runoff losses and

uptake, soil chemical properties, microbial diversity, and community composition under different fertilization and irrigation regimes. In general, the purpose of this study was to (1) verify an optimal irrigation and fertilization practice in order to minimize N and P runoff losses, and (2) explore the factors influencing N and P losses in surface runoff from paddy fields in southeastern China.

## Materials and methods

### Experiment design

Field trial was initiated in 2008 and cropped by double-cropping rice (*Oryza sativa* L.) annually at Baisha Experimental Station, Fuzhou, Fujian Province, China (26°13′31″N, 119° 04′10″E). The early and late cultivars of rice are conventional rice varieties 78–30 and 428, respectively. This region has a subtropical monsoonal climate with an average annual temperature of 19.5°C and mean annual precipitation of 1 350 mm. The soil is a typic Hapli-Stagnic Anthrosol (USDA soil system). At the beginning of the experiment, the soil had a pH (1:2.5) 6.19, 14.16 g kg$^{-1}$ soil organic matter (SOM), 0.66 g kg$^{-1}$ Total N (TN), 0.30 g kg$^{-1}$ total P (TP), 3.8 mg kg$^{-1}$ NO$_3^-$–N, 12 mg kg$^{-1}$ NH$_4^+$–N, 3.358 and 0.83 mg kg$^{-1}$ of available P (AP) and K (AK), respectively. A randomized complete block design with three treatments was conducted in 9 plots (4.0 m long × 5.0 m wide). Each treatment had three duplicates. The treatments consisted of control (no chemical fertilization with traditional flooding irrigation, T0), traditional chemical fertilization with traditional flooding irrigation (T1, based on local practices), and optimum fertilization with water-saving irrigation (T2, based on both fertilizer recommendation from local agriculture committee and water saving by shallow intermittent irrigation). The water and fertilizer practices used in this experiment are described in Table 1. The chemical compound fertilizer containing 15% N, 7% P, and 12% K was produced by China Petroleum and Chemical Co., Ltd. N, P, and K fertilizers were applied in the form of urea, superphosphate, and potassium chloride and rated according to each treatment as shown in Table 1. The proportion of N, P, and K was estimated at 46.4% of N in urea, 5% of P in calcium superphosphate, and 50% of K in potassium chloride, respectively. The 100% of the total amount of P, 60% of N, and 40% of K fertilizers were applied as basal fertilizers before planting, whereas the 40% N and 60% K fertilizers as topdressing fertilizers after tillering, respectively (Table 1). Annual fertilizer application rates were the same since 2008. Traditional flooding irrigation was needed for the rice season to be maintained at a depth of 1.0 − 6.0 cm, and water-saving irrigation at a depth of -3.0 to 3.0 cm in the paddy field. In order to prevent the exchange of water and nutrients between adjacent plots, each plot was surrounded by a concrete cement border, 40-cm deep by 30-cm width, leaving 20 cm above the soil surface for separation. At the base, a tank (2.0 m long×1.0 m wide×1.8 m deep) with vertical scale was placed to collect surface runoff beside the plot through a piping system (Fig 1). The early rice was transplanted with a 20.0 cm × 23.0 cm hill spacing on 21 April and harvested on 25 July 2018.

**Table 1. The water and fertilizer practices used in this experiment.**

| Treatment | Fertilization | Irrigation |
|:---:|:---:|:---:|
| T0 | No chemical fertilization | Traditional flooding irrigation |
| T1 | Conventional level of nitrogen (273 kg N ha$^{-1}$), phosphorus (59 kg P ha$^{-1}$), and potassium (112 kg K ha$^{-1}$) fertilizer application | Traditional flooding irrigation |
| T2 | Optimum level of nitrogen (240 kg N ha$^{-1}$), phosphorus (52 kg P ha$^{-1}$), and potassium (198 kg K ha$^{-1}$) fertilizer application | Shallow intermittent irrigation |

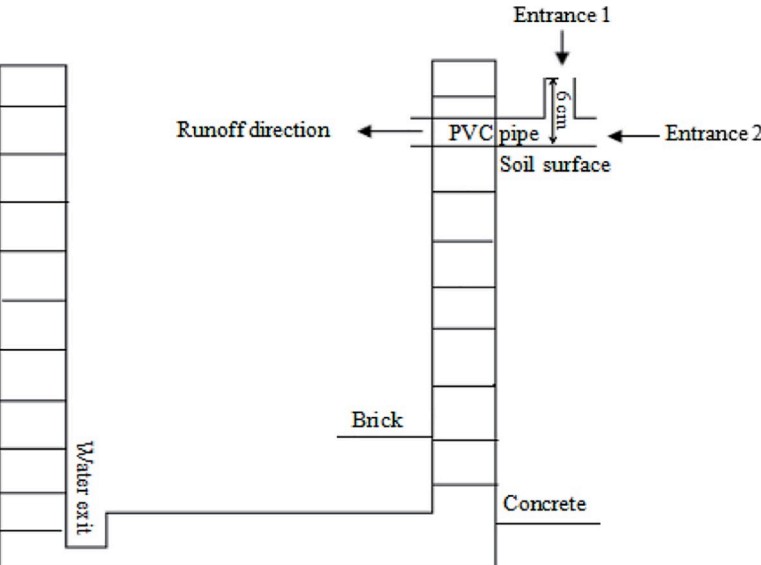

**Fig 1. Device for the collection of runoff water in experimental plots in 2018.** Notes: Entrance 1 for water into tank during irrigation period; Entrance 2 for water into tank during paddy drying or fallow period.

The late rice was transplanted with the same hill spacing on 30 July and harvested on 1 December 2018.

## Water sampling and analysis

The rainfall amount was recorded by an automatic meteorological station. After each runoff-producing-rain event, the depth in the runoff water in a tank was recorded to assess runoff volume by modelling volume—depth relationships. In addition, five runoff sub-samples of about 100 mL were collected from each plot and mixed to make a composite sample in 500 mL polyethylene bottles, then delivered on ice to the laboratory for analysis. The rest of the water was discharged to a nearby canal. The empty tanks were cleaned to prepare for the subsequent runoff collection. Prior to analysis the sample was divided into two parts. One part was filtered through a 0.45 - μm membrane to analyze $NH_4^+$–N, $NO_3^-$–N, and dissolved P (DP). The other part was filtered through a cellulose filter paper ($\leq$11 μm; ash content < 0.01%) to analyze TN and TP. TN was measured by alkaline potassium persulfate ultraviolet spectrometric method, $NO_3^-$–N was analyzed by using dual wave length ultraviolet spectrophotometric method, and $NH_4^+$–N by the indophenol blue method [20]. TP and DP were determined by the molybdate blue method after the surface water samples were digested with potassium persulfate [21]. Cumulative N (P) runoff (kg ha$^{-1}$) = sum of runoff volume (m$^3$ ha$^{-1}$) × N (P) concentration in runoff water (mg L$^{-1}$), where runoff water volume is calculated as the base area of the water tank (2.0 m × 1.0 m) × the runoff depth in the water tank.

## Plant sampling and analysis

The rice straw and grain were sampled and their yields were measured at harvest from each plot, separately (rice grain weights were adjusted to 13.5% moisture content). The rice samples were oven dried at 70°C for 72 h, weighed, and finely ground with a small ball mill for chemical analysis. Total N, P, and K in plants were determined by using the methods of diffusion, molybdenum blue colorimetry, and flame photometry, respectively [22].

## Soil sampling and analysis

**Soil physiochemical properties.** Soil samples were collected at 0−20 cm depth after late rice harvest. For each plot, five soil cores were taken and homogenized as a composite sample. One subsample was air-dried and then sieved to < 2.0 mm before physiochemical analysis. Soil moisture was calculated as the difference between oven-dry (24 h at 105°C) and fresh weight. Soil pH was measured with a glass electrode (EL20 K, Mettler-Toledo, Greifensee, Switzerland) in 1:2.5 soil:water suspension. Soil organic C (SOC) was determined by the $K_2Cr_2O_7$ oxidation-reduction titration technique. The TN content was measured spectrophotometrically after potassium persulfate digestion. Both $NH_4^+$–N and $NO_3^-$–N in 2 M KCl soil extracts (1:10 soil/extract (wt:vol)) were measured by using UV spectrophotometry. TP was measured by the alkaline fusion molybdenum-antimony colorimetric method. Olsen P in 0.5 M $NaHCO_3$ soil extracts was determined by using the molybdate blue colorimetric method [23]. The TK and AK contents were determined by flame photometry [22].

**DNA extraction and PCR amplification.** A subsample of fresh soil was stored at—80°C for molecular analysis. Total microbial DNA was extracted from 0.5 g fresh soil by using an E. Z.N.A Soil DNA Kit (Omega Bio-tek, Norcross, Georgia, USA), according to the manufacturer's protocol [24]. A NanoDrop-2000 Spectrophotometer (Thermo Fisher Scientific, Waltham, MA, USA) was used to determine the purities and concentrations of extracted DNA, and the V3 − V4 region of the bacterial 16S rRNA gene was amplified by PCR (95°C for 3 min, followed by 25 cycles at 95°C for 30 s, 55°C for 30 s, 72°C for 45 s and a final extension at 72°C for 10 min) by using the forward primer 338F (5'-barcode- `ACTCCTACGGGAGGCAGCA` -3') and the reverse primer 806R (5'-`GGACTACHVGGGTWTCTAAT`-3'), where a barcode is an unique eight-base sequence for each sample [25]. Meanwhile, The V4 − V5 region in the 18S ribosomal RNA gene of the fungi was amplified by PCR (95°C for 3 min, followed by 25 cycles at 95°C for 30 s, 55°C for 30 s, and 72°C for 45 s and a final extension at 72°C for 10 min) using primers SSU0817F 5'-barcode- `TTAGCATGGAATAATRRAATAGGA`)-3' and 1196R 5'-`TCTGGACCTGGTGAGTTTCC`-3', where a barcode is an unique eight-base sequence for each sample [26]. The PCR mixture (20 μL) contained 4 μL 5 × FastPfu Buffer, 2 μL 2.5 mmol $L^{-1}$ dNTPs, 0.8 μL each primer (5 μmol $L^{-1}$), 0.4 μL FastPfu Polymerase, and 10 ng template DNA [26].

**Illumina MiSeq sequencing.** The amplified DNA was subjected to horizontal electrophoresis on 2% agarose and purified with an AxyPrep DNA Gel Extraction Kit (Axygen Biosciences, Union City, California, USA) according to the manufacturer's instructions and quantified by using QuantiFluo-ST (Promega, Madison, Wisconsin, USA). The purified amplicons were pooled in equimolar concentrations and paired-end sequenced (2 × 250) on an Illumina MiSeq platform according to the standard protocols [27]. The raw reads were deposited into the National Center for Biotechnology Information (NCBI) Sequence Read Archive (SRA) database with accession number SRP293735.

**Illumina data analysis.** Raw fastq files were demultiplexed, quality-filtered, and analysed by using Quantitative Insights Into Microbial Ecology (QIIME) 1.17 [28]. These sequences were clustered into operational taxonomic units (OTUs) at 97% sequence similarity by the UPARSE pipeline (version 7.0.1090) [29]. Using the UPARSE (version 7.0.1090), we also removed singleton sequences (i.e., sequences appearing only one time in the entire data set). In addition, chimeric sequences were identified and removed by using UCHIME. The taxonomy of 16S and 18S rRNA gene sequences was analyzed by RDP Classifier (http://rdp.cme.msu.edu/) against the Silva rRNA database (version 1.30.2) using a confidence threshold of 70% [30]. As the number of sequence reads in each sample varied, the OTU table was rarified (holding the same sequence number in each sample) prior to microbial community diversity

calculations. Rarefaction curves and other OTUs-based analyses such as the abundance-based coverage estimators (ACE) and Chao1, Shannon-Wiener index (H′), and Simpson's index (D) were conducted by the mothur software package (version 7.0.1090) [25]. Chao1 and ACE were calculated to estimate the richness of microbial community based on sequence dissimilarity. The diversity within each sample was estimated by H′ and D [31].

## Data analysis

Statistical analyses were done by using SAS software, version 8.02 (SAS Institute Inc., Carey, North Carolina, USA). All values were expressed as means ± SD ($n$ = 3). The one-way analysis of variance and the Duncan multiple—range test were applied to determine the differences in N and P runoff losses, uptake, microbial diversity, edaphic characteristics, and rice yields at three water and fertilizer regimes in 2018. To better compare microbial community similarities, partial least squares discriminant analysis (PLS—DA) was performed by PLS regression methods. In addition, the similarities and differences among microbial communities were also described by using the number of shared and unique OTUs in the three treatments by a Venn diagram. To compare the top 10 microbial genera, a heatmap analysis was performed, and the result was plotted in Vegan packages in R software (version 2.15.3) [32]. Furthermore, a heatmap of correlations between the relative abundances of microbial taxa and edaphic characteristics (e.g., pH, SOC, and TN) was tested by using the Canoco software for Windows Version 4.5 [33]. In addition, environmental factors were selected by the functions of envfit (permu = 999) and variance inflation factor (vif).cca, and the environmental factors with vif > 10 were removed from the following analysis. The vif values of SOC, $NH_4^+$–N, $NO_3^-$–N, TP, and AK were higher than 10 and removed. Additionally, Pearson correlations were performed between the microbial abundances and N and P runoff losses. The unweighted UniFrac distance—based redundancy analysis (db-RDA) was processed by R software (version 2.15.3) to determine which soil variables were related to soil microbial community structures [32]. Additionally, Pearson correlations were performed between the microbial abundances and N and P runoff losses. Furthermore, RDA was selected, depending on the length of gradient calculated by detrended correspondence analysis (DCA). In this study, the gradient length was smaller than 3.0, so RDA was chosen to analyze the correlations between soil N and P runoff losses and their impact factors [34]. The influencing factors included the runoff volume, fertilizer inputs, and soil chemical properties. The method of the rank analysis was performed by using Canoco for Window 4.5.

## Results

### Rice yields and soil fertility

The T1 and T2 treatments increased grain yield by 65.9% and 90.4%, respectively, compared to the T0 treatment in the early rice season, while increased grain yield by 91.9% and 93.0% compared to the T0 treatment in the late rice season (Table 2). However, there were no significant differences in the grain yield between T1 and T2 treatments. In addition, $K^+$ uptake in rice plants of T1 and T2 treatments in the late rice season was about 1.37 and 1.06 times higher than that in the early rice season, respectively (Table 2). Meanwhile, the T2 treatment had higher contents of soil pH, SOC, TK and AK than those in the T1 treatment. Nevertheless, soil $NO_3^-$–N content significantly ($P < 0.05$) increased in the T1 treatment but decreased in the T2 treatment as compared to that of the T0 treatment. Additionally, all three treated paddy soils were acidic (Table 2).

**Table 2. Soil properties and plant traits as influenced by fertilization and irrigation in 2018.**

| Treatment | Soil properties | | | | | | | | | Plant traits | | | | | | | |
|---|---|---|---|---|---|---|---|---|---|---|---|---|---|---|---|---|---|
| | pH | SOC | TN | TP | TK | $NO_3^--N$ | $NH_4^+-N$ | Olsen−P | AK | Early rice | | | | Late rice | | | |
| | | | | | | | | | | Grain yield | N | P | K | Grain yield | N | P | K |
| | | g kg⁻¹ | | | | mg kg⁻¹ | | | | kg ha⁻¹ | g kg⁻¹ | | | kg ha⁻¹ | g kg⁻¹ | | |
| T0 | 5.97 ±0.08b | 15.16 ±0.10b | 2.18 ±0.25a | 0.29 ±0.01c | 20.31 ±0.73ab | 13.20 ±0.77b | 35.14 ±6.12b | 0.96 ±0.06c | 64.99 ±4.62b | 2971 ±374b | 81.64 ±4.03b | 14.14 ± 0.84b | 56.96± 3.08a | 2795 ±165b | 83.95 ±1.66c | 10.60 ±0.45b | 52.76 ±2.19c |
| T1 | 6.01 ±0.08b | 15.33 ±0.13b | 2.00 ±0.12a | 0.41 ±0.01b | 19.68 ±0.59b | 18.74 ±2.94a | 59.64 ±5.53a | 2.75 ±0.19a | 57.87 ±3.38b | 4929 ±518a | 96.88 ±4.27a | 18.84 ± 0.68a | 45.40 ±1.73b | 5363 ±119a | 108.16 ± 0.83a | 21.36± 0.40a | 107.80 ± 7.64b |
| T2 | 6.24 ±0.11a | 15.98 ±0.18a | 1.84 ±0.16a | 0.48 ±0.02a | 21.55 ±0.55a | 6.37 ±0.96c | 51.55 ±7.43a | 2.41 ±0.17b | 86.15 ±6.17a | 5656 ±134a | 88.39 ±2.74b | 20.47 ± 0.55a | 60.81 ±1.82a | 5393 ±105a | 101.04 ±2.57b | 20.62 ±1.56a | 125.14 ±4.49a |

Notes: T0 = Traditional irrigation; T1 = Traditional irrigation and fertilization practice; T2 = Water-saving irrigation and optimizing fertilization. SOC: Soil organic carbon; N: Nitrogen; TN: Total N; $NO_3^--N$: Nitrate N; $NH_4^+-N$: Ammonium N; P: Phosphorus; TP: Total P; K: Potassium; TK: Total K; AK: Available K. Values (means ± SD) with different lower-case letters in a column are significantly different at $P < 0.05$ according to the Duncan test.

## Nitrogen and phosphorus losses

A total of 17 runoff-producing rainfall events were recorded during the rice growing season from 1 May to 9 September 2018, and they ranged from 7.0 to 101.7 mm. Among them, three extreme precipitation events with a daily rainfall greater than 60.0 mm were observed, 67.0 mm on June 21, 89.0 mm on July 10, and 101.7 mm on September 6 (Fig 2). High runoff fluxes

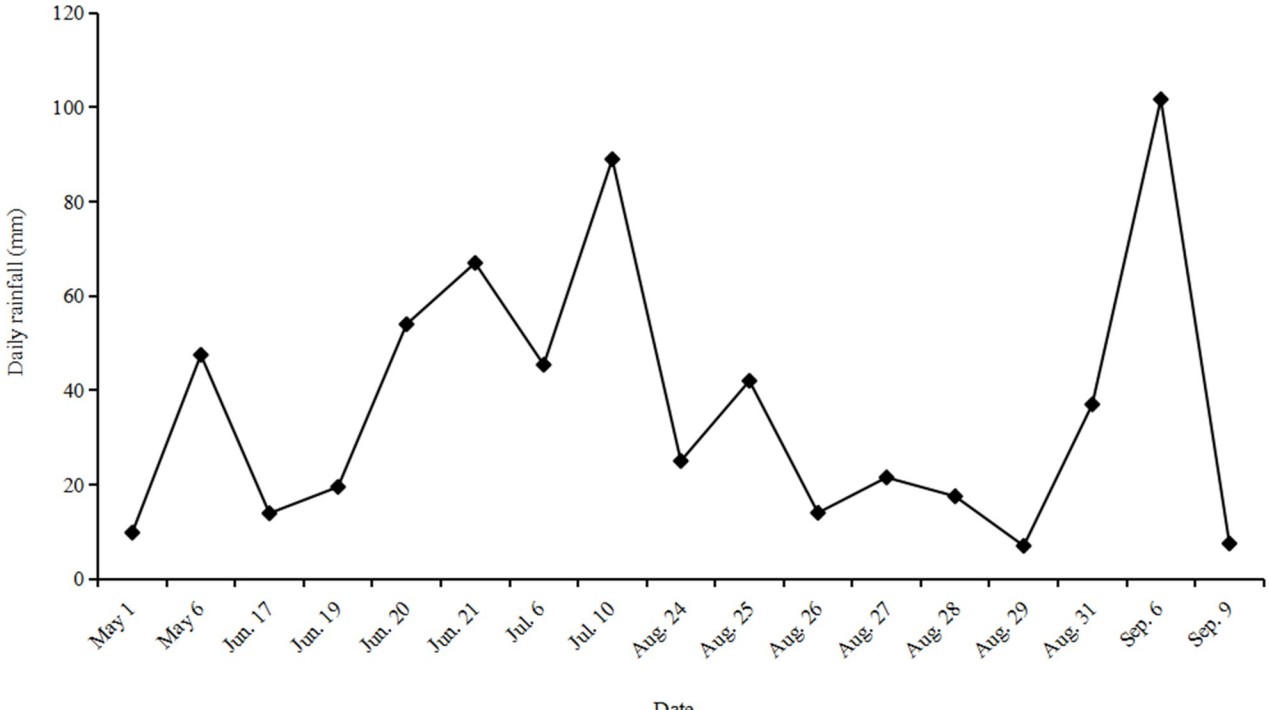

**Fig 2. Characteristics of 17 rainfall-runoff events in the experiment plots from May to September 2018.**

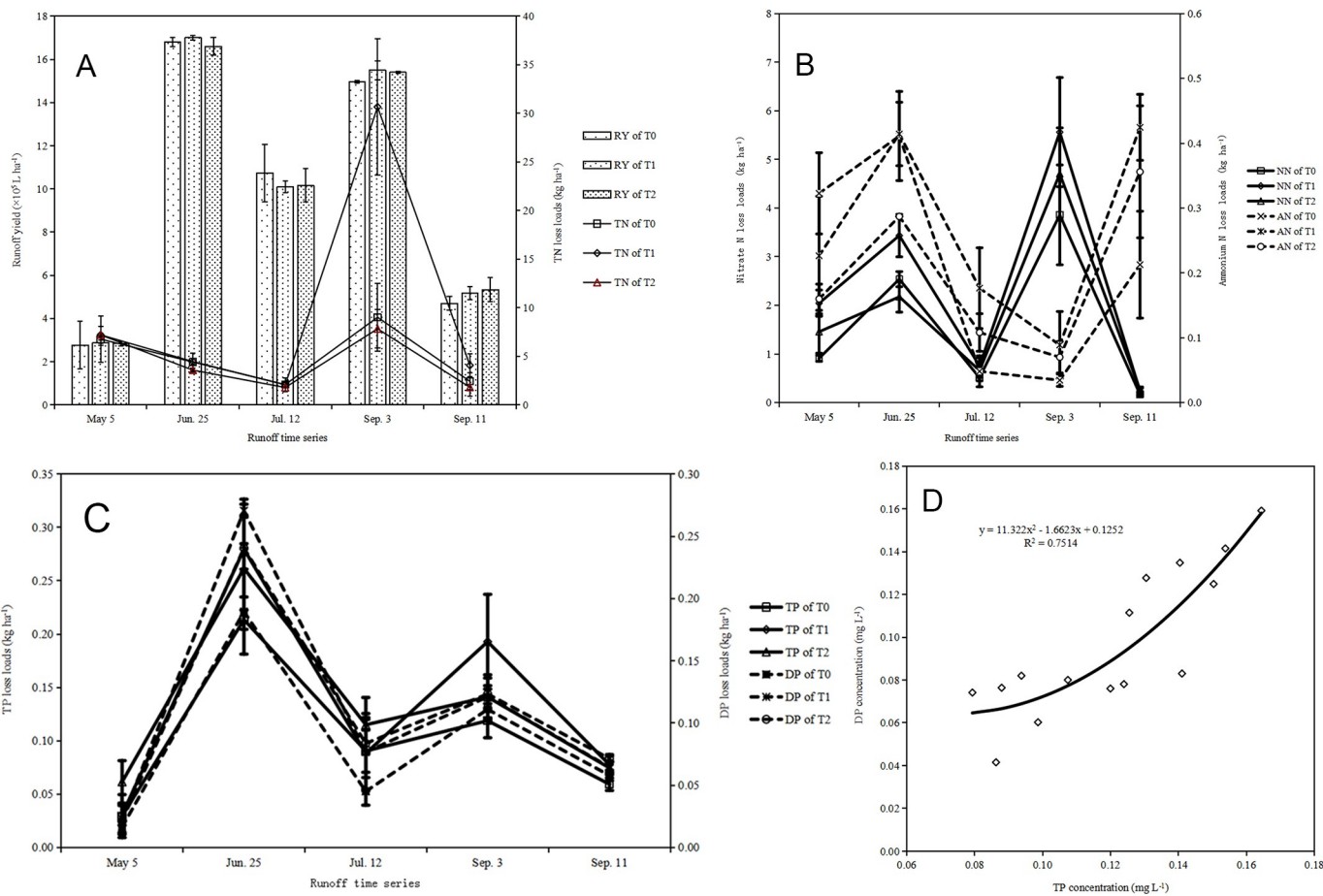

**Fig 3.** Runoff Nitrogen (N) and phosphorus (P) losses from rice fields as influenced by the treatments T0, T1, and T2 from January to December 2018: Accumulated TN losses (A), AN and NN concentrations (B), accumulated TP and DP losses (C), and relationship between TP and DP concentrations (D). Notes: T0 = Traditional irrigation; T1 = Traditional irrigation and fertilization practice; T2 = Water-saving irrigation and optimizing fertilization. RY: Runoff yields. TN: Total N; NN: Nitrate N; AN: Ammonium N; TP: Total P; DP: Dissolved P.

of surface flow generally occurred from May to September 2018, when facing the high precipitation (Figs 2 and 3A). Moreover, the runoff flux had a close relationship with $NO_3^-$–N, TP and DP losses in all treatments (Fig 3A–3C). Meanwhile, the loss ratio of N was higher than that of P. In addition, the loss ratio of N from surface runoff in the T1 plots was the highest among the treated plots. By contrast, the T2 treatment reduced N loss from paddy fields by 21.21 kg N ha$^{-1}$, especially because the N—fertilizer use efficiency was high in the T2 treatment in comparison with that in the T1 treatment (Table 3). Inorganic dissolved N loss accounted for 29.57–47.05% of N loss under different irrigation and fertilization regimes, and a greater proportion of the loss was in the $NO_3^-$–N form (Table 3). $NO_3^-$–N loss in the late rice season were higher than that in the early rice season (Fig 3B). Additionally, $NO_3^-$–N loss from the T1 treatment was significantly ($P < 0.05$) higher than that of the other treatments. Nevertheless, no significant difference was found in $NH_4^+$–N loss among three treatments (Table 3). Compared to the T0 treatment, the T1 and T2 treatments increased P loss by 30.92% and 33.07%, respectively (Table 3). However, the T2 treatment did not lead to significantly ($P < 0.05$) different P loss compared to the T1 treatment. DP was the major form of P loss, and accounted for 82.19%, 82.96%, and 79.41% for TP loss in T0, T1 and T2, respectively (Table 3). Moreover, DP loss was positively correlated with TP loss (Fig 3D). As the fertilizer level increased, the N

**Table 3. Annual loads of nitrogen and phosphorus transported by surface runoff for the treatments T0, T1, and T2 from January to December 2018.**

| Treatment | Runoff (×10⁵ L ha⁻¹) | Fertilizer amount (kg ha⁻¹) | | Nitrogen and phosphorus losses in runoff (kg ha⁻¹) | | | | | Loss ratio (%) | | AEN (kg kg⁻¹ N) | AEP (kg kg⁻¹ P) |
|---|---|---|---|---|---|---|---|---|---|---|---|---|
| | | TN | TP | TN | NO₃⁻-N | NH₄⁺-N | TP | DP | TN | TP | | |
| T0 | 50.0±0.6a | 0 | 0 | 24.67±2.40b | 7.77±1.20b | 1.03±0.13a | 0.51±0.03b | 0.42±0.02b | — | — | — | — |
| T1 | 50.6±0.3a | 273 | 59 | 43.39 ±14.04a | 11.63 ±1.15a | 1.20±0.29a | 0.67±0.04a | 0.56±0.03a | 15.89±5.14a | 1.13±0.08a | 16.58±0.42b | 76.72±1.94b |
| T2 | 50.4±1.1a | 240 | 52 | 22.17 ± 1.07b | 9.25 ± 0.94b | 1.18 ±0.32a | 0.68 ± 0.08a | 0.54 ± 0.08a | 9.24±0.45b | 1.31±0.16a | 22.02±2.38a | 101.61±11.00a |

Notes: T0 = Traditional irrigation; T1 = Traditional irrigation and fertilization practice; T2 = Water-saving irrigation and optimizing fertilization. N: Nitrogen; P: Phosphorus; Loss ratio, total nitrogen (phosphorus) loss loading/amount of applied nitrogen (phosphorus); AEN (P), agronomic N (P) use efficiency, increased grain yield/unit N (P) application. TN: Total N; NO₃⁻−N: Nitrate N; NH₄⁺−N: Ammonium N; TP: Total P; DP: Dissolved P. Values (means ± SD) with different lower-case letters in a column are significantly different at $P < 0.05$ according to the Duncan test.

and P runoff losses showed an upward trend (Fig 3A–3C). The cumulative P loss from paddy fields in the early rice season was greater than that in the late rice season in all treatments (Fig 3C). The highest P loss was also observed in all treatments on June 25 in the early rice season (Fig 3C).

## Microbial alpha diversity

High query coverage (>98.0%) suggested that this study captured the dominant OTUs of microbia in each soil sample (Table 4). Moreover, all of the rarefaction curves of bacterial 16S rRNA and fungal 18S rRNA sequences in soil samples reached saturation, suggesting that the number of sequence reads was sufficient to represent most of sequence types (S1A and S1B Fig). The numbers of 16S rRNA OTUs from bacteria at a 97% sequence identity were 1973, 2055, and 2018 as well as 266, 248, and 250 for fungal OTUs in soil samples in the T0, T1 and T2 treatments, respectively (Fig 4A and 4B). Most bacterial OTUs (85.79%) were shared (Fig 4A), while 179 of 323 fungal OTUs were shared among three treated soil samples (Fig 4B). Meanwhile, we also found that many of the alpha diversity indices were nonsignificantly (P>0.05) different among three treatments (Table 4). No variations in the soil microbial alpha diversity (except Chao 1) among different treatments may be explained by their response to natural mechanisms rather than by direct impacts of fertilization and irrigation treatments on bacteria and fungi. Additionally, bacterial alpha diversity indices (ACE, Chao 1 and Shannon-Wiener index) were significantly (P < 0.05) higher than those of the fungi in the paddy soil treated with different irrigation and fertilization strategies (Table 4).

**Table 4. Microbial alpha diversity as affected by fertilization and irrigation in 2018.**

| Microbe | Treatment | Coverage (%) | Richness | | Diversity | |
|---|---|---|---|---|---|---|
| | | | ACE | Chao 1 | H′ | D (×10⁻³) |
| Bacteria | T0 | 98.71±0.07a(b) | 1789±61a(a) | 1812±37b(a) | 6.23±0.15a(a) | 5.57±0.78a(a) |
| | T1 | 98.77±0.02a(b) | 1885±23a(a) | 1892±26a(a) | 6.41±0.11a(a) | 4.62±1.16a(b) |
| | T2 | 98.72±0.05a(b) | 1841±47a(a) | 1858±48ab(a) | 6.33±0.11a(a) | 5.53±2.06a(b) |
| Fungi | T0 | 99.97±0.00a(a) | 166±32a(b) | 167±34b(b) | 3.00±0.48a(b) | 137.13±95.62a(a) |
| | T1 | 99.96±0.00a(a) | 171±14a(b) | 173±18a(b) | 3.20±0.36a(b) | 86.43±43.58a(a) |
| | T2 | 99.95±0.00a(a) | 199±8a(b) | 201±6ab(b) | 3.15±0.23a(b) | 95.38±51.71a(a) |

Notes: T0 = Traditional irrigation; T1 = Traditional irrigation and fertilization practice; T2 = Water-saving irrigation and optimizing fertilization. Operational taxonomic units (OTUs); Abundance-based coverage estimators (ACE); H′: Shannon-Wiener index; D: Simpson's index. Values (means ± SD) with different lower-case letters inside and outside the parentheses in a column are significantly different between soil microbes or fertilizer treatments at $P < 0.05$ according to the Duncan test.

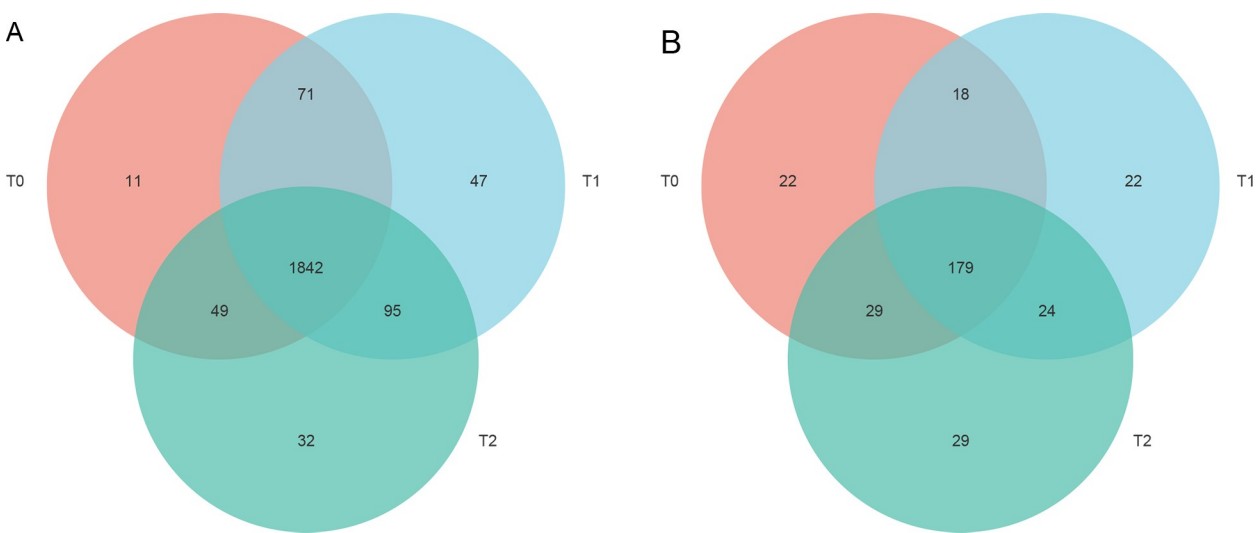

**Fig 4.** Venn diagram depicts bacterial (A) and fungal (B) operational taxonomic units (OTUs) that were shared or unique for T0, T1, and T2. Notes: T0 = Traditional irrigation; T1 = Traditional irrigation and fertilization practice; T2 = Water-saving irrigation and optimizing fertilization.

## Microbial community composition

Each water and fertilizer treatment formed a unique microbial community structure by PLS—DA approach (Fig 5A and 5B). A total 34.85% of the variations in the composition of bacterial communities could be explained by the first two principal components, and a total 25.94% of the variations in the composition of fungal communities by the first two principal components (Fig 5A and 5B). Moreover, T2 increased the relative abundances of the bacterial phyla *Actinobacteria*, *Cyanobacteria*, and *Verrucomicrobia* compared to those in other treatments. Nevertheless, the T2 treatment decreased the abundance of *Acidobacteria* by 12.19% and 19.88%, respectively, compared to that in the T0 and T1 treatments (Fig 6A). Moreover, the predominant bacterial phyla in paddy soils were *Proteobacteria* (the number of classified sequences in

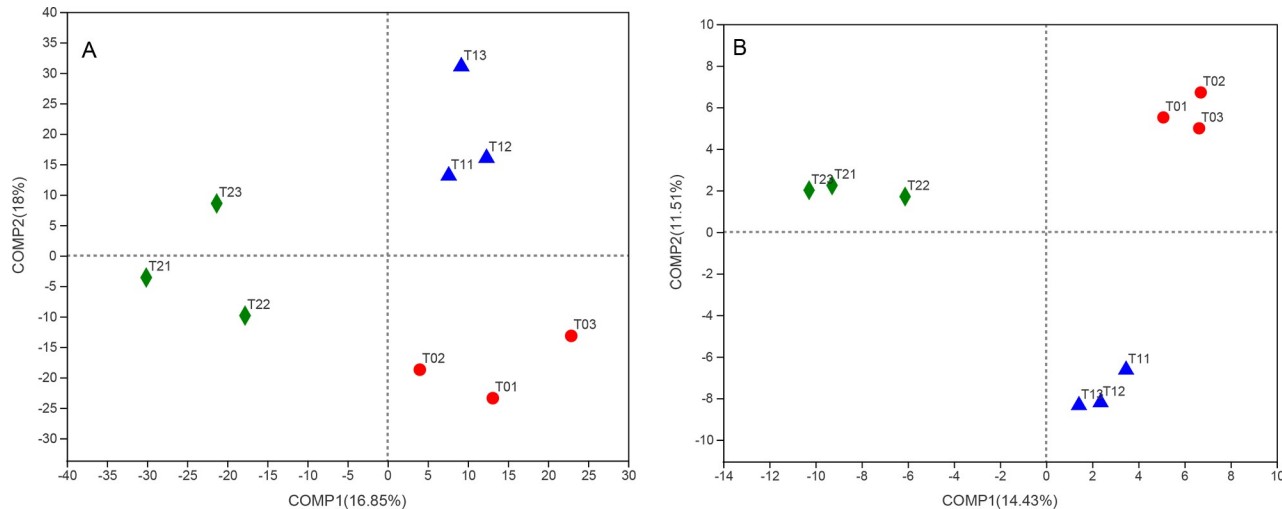

**Fig 5.** Partial least squares discriminant analysis (PLS—DA) is an adaptation of PLS regression methods to represent differences in the community structure of bacterial (A) and fungal (B) microbiota that was associated with T0, T1, and T2. Notes: T0 = Traditional irrigation; T1 = Traditional irrigation and fertilization practice; T2 = Water-saving irrigation and optimizing fertilization.

this phylum ranged from 28.02 to 32.97% in all the samples), *Chloroflexi* (23.43–30.54%), and *Acidobacteria* (9.51–11.87%); the rare phyla were characterized by low *Cyanobacteria*, *Bacteroidetes*, *Gemmatimonadetes*, and *Verrucomicrobia* abundances (Fig 6A). Additionally, the classified sequences from each treated soil were affiliated with the fungal phyla: *Ascomycota*, *Basidiomycota*, *Mucoromycota*, and *Chytridiomycota*; the remaining sequences were unclassified fungi and other classified fungal phyla (Fig 6B). *Ascomycota* and *Basidiomycota* were the two most abundant fungal phyla in soils under different water and fertilizer treatments. Moreover, T2 increased the abundance of *Mucoromycota* by 30.72%, whereas T1 decreased *Mucoromycota* by 2.94% in comparison with that of T0 (Fig 6B).

## Factors impacting N and P surface runoff losses

Soil pH ($r^2$ = 0.8973, $P$ = 0.003) and Olsen P content ($r^2$ = 0.6609, $P$ = 0.033) were significantly correlated with bacterial community structure by a db—RDA (Fig 7A). Meanwhile, soil pH ($r^2$ = 0.8123, $P$ = 0.007) and TN content ($r^2$ = 0.67599, $P$ = 0.024) were significantly correlated with fungal community structure (Fig 7B). In addition, the relative abundance of *Desulfobacca* bacteria was significantly ($P$ < 0.05) positively related to SOC and Olsen P contents, whereas the relative abundance of *Nitrospira* bacteria was significantly ($P$ < 0.001) negatively related to soil AK content (Fig 8A). The relative abundance of *Leucosporidium* fungi was significantly negatively related to soil AK content but positively correlated with soil Olsen P content ($P$ < 0.05) (Fig 8B). Taken together, soil properties could alter microbial community composition, which was also the crucial contributing factor for N and P runoff losses under different fertilization and irrigation regimes. N and P losses in runoff were positively correlated with the relative abundances of the bacterial phyla *Firmicutes*, *Bacteroidetes*, and *Gemmatimonadetes* and the fungal phyla *Ascomycota*, whereas the nutrient runoff losses were negatively correlated with the abundances of the bacterial phyla *Chloroflexi* and the fungal phyla *Basidiomycota* and *Chytridiomycota* (Table 5). Meanwhile, the losses of TN and $NO_3^-$–N in the runoff were positively related to the abundances of the bacterial phyla *Proteobacteria* and *Bacteroidetes*, but negatively to the abundances of the bacterial phyla *Planotomycetes* and *Verrucomicrobia* (Table 5). Meanwhile, there was a significant ($P$ < 0.05) and positive relationship between the abundance of *Nitrospirae* bacteria and TN runoff loss. In contrast, a negative correlation occurred between the abundance of *Mucoromycota* fungi and TN runoff loss (Table 5). In addition, there existed a positive association of TP and DP losses in the runoff with the

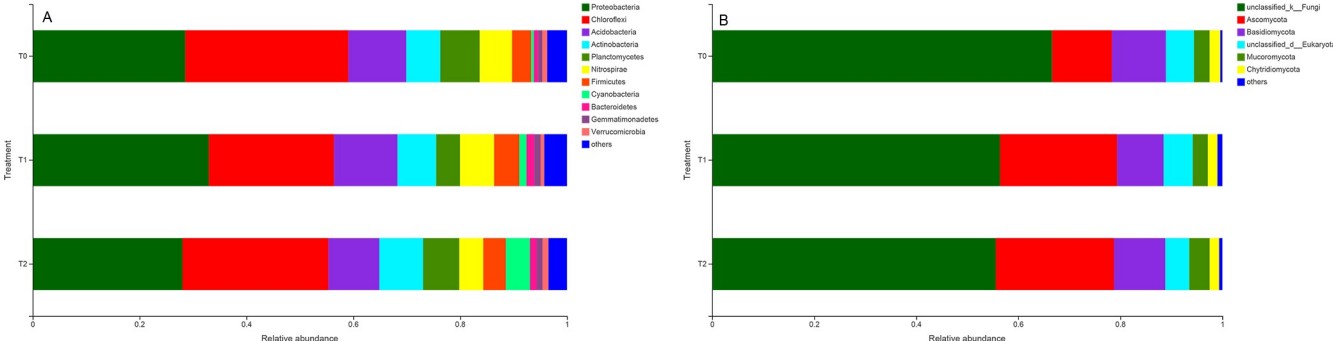

**Fig 6.** Average relative abundance of dominant bacterial (A) and fungal (B) phyla (> 1.0%) in different fertilization and irrigation regimes. The abundance is expressed as the average percentage of the targeted sequences to the total high-quality bacterial and fungal sequences of samples from triplicate plots of each fertilization regime, respectively. Notes: 'Others' refer to those identified phyla with lower than 1.0% relative abundance in all the samples. T0 = Traditional irrigation; T1 = Traditional irrigation and fertilization practice; T2 = Water-saving irrigation and optimizing fertilization.

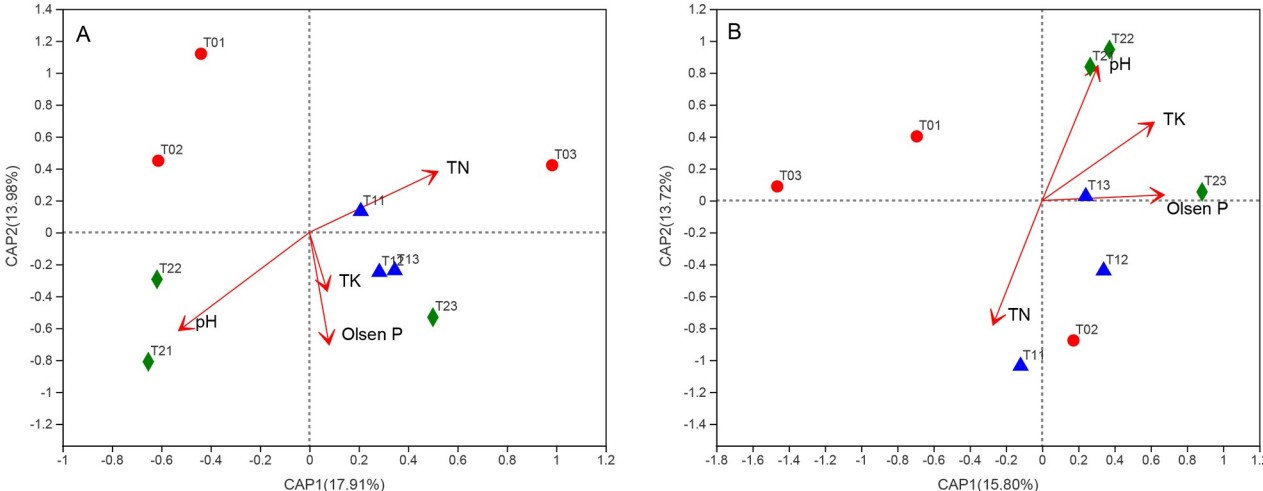

**Fig 7.** Distance-based redundancy analysis (db-RDA) of the bacterial (A) and fungal (B) communities based on environmental factors. Notes: T0 = Traditional irrigation; T1 = Traditional irrigation and fertilization practice; T2 = Water-saving irrigation and optimizing fertilization. N: Nitrogen; TN: Total N; P: Phosphorus; K: Potassium; TK: Total K.

abundance of *Actinobacteria* bacteria (Table 5). Meanwhile, the positive association of TP loss with the abundance of *Cyanobacteria* was found (Table 5). In addition, all the selected environmental factors interpreted the majority of N loss variations (94.5%), and P loss variations were totally interpreted by the environmental factors by using RDA (Fig 9A and 9B). Moreover, the N loss via surface runoff was mainly due to high N fertilizer input, soil $NO_3^-$ –N, and $NH_4^+$ –N content, whereas the P loss largely depended on high P fertilizer input, soil TP, and Olsen-P content (Fig 9A and 9B).

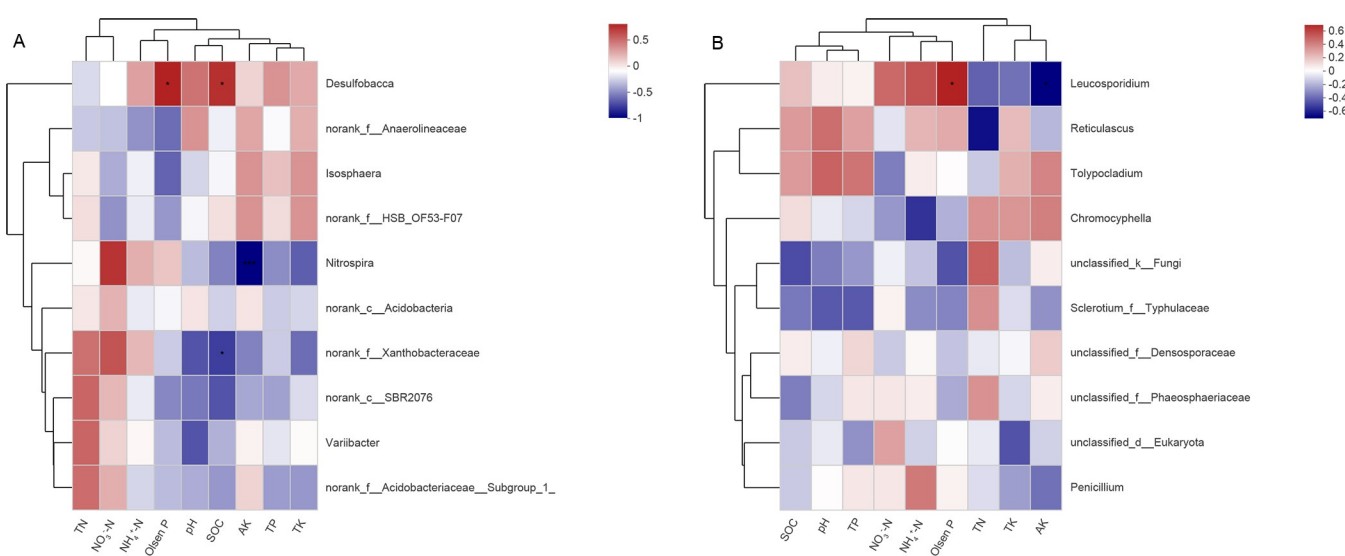

**Fig 8.** Correlation heatmap of soil properties and relative abundances of bacterial (A) and fungal (B) communities at the genus level. $^*0.01 < P \leq 0.05$; $^{**}0.001 < P \leq 0.01$; $^{***}P \leq 0.001$. Notes: T0 = Traditional irrigation; T1 = Traditional irrigation and fertilization practice; T2 = Water-saving irrigation and optimizing fertilization. SOC: Soil organic carbon; N: Nitrogen; TN: Total N; $NO_3^-$–N: Nitrate N; $NH_4^+$–N: Ammonium N; P: Phosphorus; TP: Total P; K: Potassium; TK: Total K; AK: Available K.

**Table 5. Correlations of runoff losses of nitrogen and phosphorus and the abundances of bacteria and fungi.**

| Taxon | Nutrient runoff losses | | | | |
|---|---|---|---|---|---|
| | TN | $NO_3^-$-N | $NH_4^+$-N | TP | DP |
| Bacteria | | | | | |
| Proteobacteria | 0.9999** | 0.8937** | 0.5450 | 0.3595 | 0.5034 |
| Chloroflexi | -0.8407** | -0.9972** | -0.9080** | -0.8015** | -0.8864** |
| Acidobacteria | 0.8851** | 0.5558 | 0.0819 | -0.1263 | 0.0331 |
| Actinobacteria | -0.1358 | 0.3532 | 0.7633* | 0.8805** | 0.7939* |
| Planotomycetes | -0.9508** | -0.9834** | -0.7715* | -0.6230 | -0.7395* |
| Nitrospirae | 0.7163* | 0.2967 | -0.2047 | -0.4030 | -0.2523 |
| Firmicutes | 0.7606* | 0.9782** | 0.9559** | 0.8744** | 0.9404** |
| Cyanobacteria | -0.4201 | 0.0637 | 0.5405 | 0.7030* | 0.5810 |
| Bacteroidetes | 0.6250 | 0.9217** | 0.9941** | 0.9500** | 0.9876** |
| Gemmatimonadetes | 0.6748* | 0.9451** | 0.9848** | 0.9275** | 0.9752** |
| Verrucomicrobia | -0.9398** | -0.6630* | -0.2157 | -0.0088 | -0.1678 |
| Fungi | | | | | |
| Ascomycota | 0.3885 | 0.7810* | 0.9861** | 0.9991** | 0.9930** |
| Basidiomycota | -0.8954** | -0.9994** | -0.8561** | -0.7305* | -0.8298** |
| Mucoromycota | -0.6521 | -0.2115 | 0.2900 | 0.4819 | 0.3364 |
| Chytridiomycota | -0.3341* | -0.7432* | -0.9747** | -0.9999** | -0.9844** |

Notes: T0 = Traditional irrigation; T1 = Traditional irrigation and fertilization practice; T2 = Water-saving irrigation and optimizing fertilization. N: Nitrogen; TN: Total N; $NO_3^-$-N: Nitrate N; $NH_4^+$-N: Ammonium N; P: Phosphorus; TP: Total P; DP: Dissolved phosphorus.

*, **Significant at the 0.05 and 0.01 probability level, respectively.

## Discussion

### Effects of different water and fertilizer treatments on nitrogen and phosphorus losses

A total of 17 runoff-producing rainfall events occurred during the experimental period. Among them, three precipitation events with values over 60 mm were extreme events (Fig 2). Furthermore, a significant positive relationship between rainfall and runoff was reported for rice system [35]. Runoff DN loss was mainly in the form of $NO_3^-$–N than in $NH_4^+$–N in different water and fertilizer treatments (Table 3). The result is consistent with the previous report in which the N loss by surface runoff was mainly in the form of $NO_3^-$–N, respectively, from vegetable, upland crop, and rice systems under natural rainfall [35]. These results indicated that $NO_3^-$–N was the major form of N in the surface runoff. The reason was that $NH_4^+$–N was more easily adsorbed by soil colloidal particles than $NO_3^-$–N resulting in the slow migration of $NH_4^+$–N in soils [36], and $NH_4^+$–N could also be converted into $NO_3^-$–N by nitrification; hence, this would contribute to the preferential loss of easily mobile $NO_3^-$–N during successive rainfall events [37]. However, the loss ratio of P was only 1.13−1.31% (Table 3). The finding agreed with that obtained by Yi *et al.* (2018), who found that the loss ratio of P in surface runoff was lower than 1% [14]. The small amount of P runoff loss was mainly due to the studied soils, typic hapli—stagnic anthrosols, their enrichment of Fe and Al oxides, which was helpful to adsorb additional P resulting in less runoff loss of P from the paddy field [38]. Moreover, DP was the main P loss in the runoff in different water and fertilizer treatments (Table 3).

High precipitation also caused large fluxes of DP, TP and $NO_3^-$–N in all treatments (Fig 3A–3C), indicating that rainfall was another risk factor for the increasing nutrient runoff losses, which was similar to that reported in other studies [14]. Additionally, DP loss was

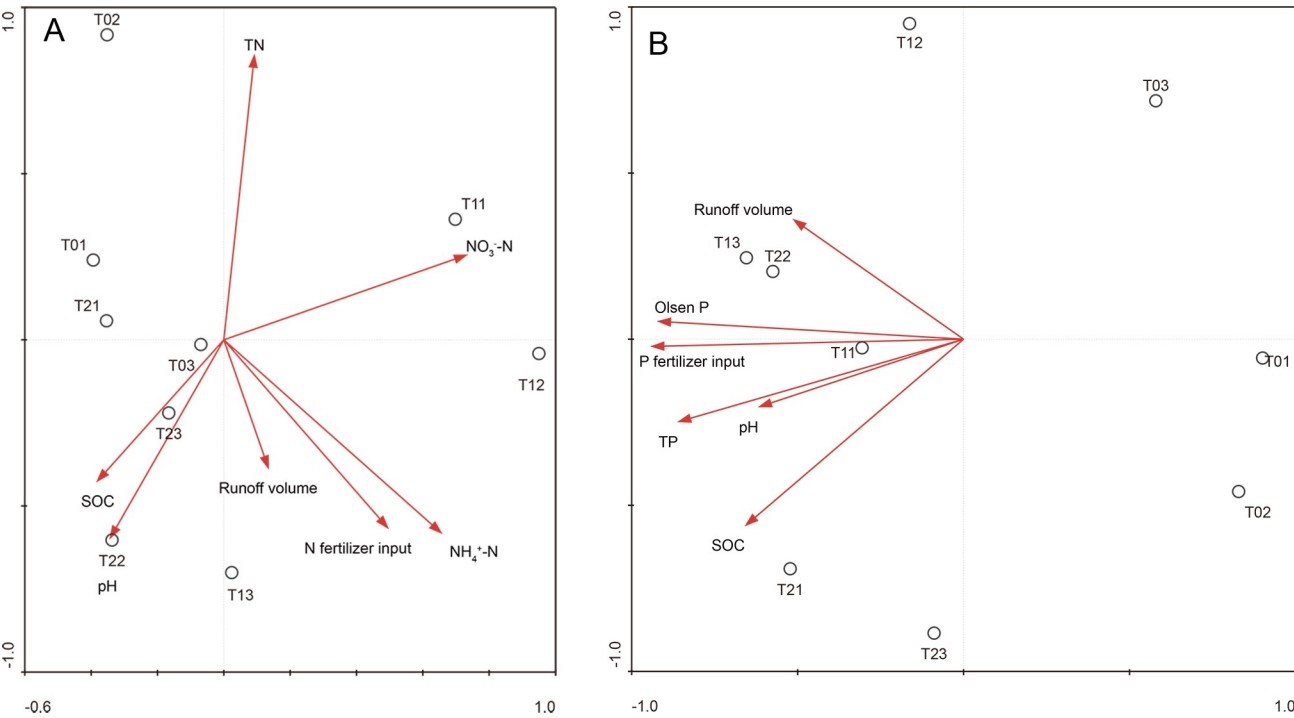

**Fig 9.** The impact factors determining N/P runoff losses by redundancy analysis (A)/(B). Notes: T0 = Traditional irrigation; T1 = Traditional irrigation and fertilization practice; T2 = Water-saving irrigation and optimizing fertilization. SOC: Soil organic carbon; N: Nitrogen; TN: Total N; $NO_3^-$−N: Nitrate N; $NH_4^+$−N: Ammonium N; P: Phosphorus; TP: Total P.

highly positively correlated with TP loss (Fig 3D). This result is consistent with that of Zhao *et al.* (2017), who indicated that the TP and AP concentrations in runoff had a strong correlation ($R^2$ = 0.933), mainly because the P in the runoff water was mainly in the form of available P [39]. Consequently, DP runoff can be used to estimate TP runoff. Nevertheless, Liu *et al.* (2020) suggested that PP (particulate phosphorus) was mainly moved via surface flow, accounting for 69.4–79.7% of TP in a double rice-cropping system in the subtropical hilly region of China [10]. The process depended mainly on the paddy water, which had the strong adsorption of PP due to its high organic matter and clay contents [40]. In this study, the P loss in the early rice season was higher than that in the late rice season (Figs 2 and 3C). In addition, the average loss of P fractions in the surface runoff was lower than that of N fractions (Fig 3A–3C). Similar results were reported by Huang *et al.* (2020), who showed that the average runoff loss of TP and DP was lower than those of TN and DN in all treatments [7]. The results indicated that the risk of N loss in surface runoff was higher than that of P loss in the double rice cropping system in the subtropical region of China. Moreover, the N rather than P runoff losses in the T1 treatment significantly ($P < 0.05$) increased compared to those in the T2 treatment (Table 3). This result was consistent with the finding of Liu *et al.* (2020), suggesting that greater amounts of N and P fertilizers resulted in more substantial N loss through surface runoff from a paddy field [10]. That was because excessive N fertilizer applications to the intensive rice systems resulted in a large amounts of N accumulated in the paddy soil, consequently created a large soil N pool, which contributed to the preferential loss of easily mobile N runoff loss during successive rainfall events compared with the immobile and occluded P in rice paddy soils [41,42]. Furthermore, the greater N input led to a decrease in soil pH and thus enhanced P accumulation in the soil [43]. Overall, the optimal fertilization and irrigation for rice could reduce the N runoff loss from paddy fields.

## Effects of different water and fertilizer treatments on soil microbial community

Numerous diversity indices, including species richness and evenness together, are also called heterogeneity indices. In this study, the acid pH-range in different water and fertilizer treatments was 5.97–6.24, but the pH changes did not result in alterations in microbial alpha diversity (except Chao 1) (Table 4). No variations in the soil microbial alpha diversity among the different treatments may be explained by their response to various natural and specific conditions (e.g., climatic factor and floristic composition) on microbes [44,45]. However, Joa *et al.* (2014) showed that soil pH was significantly ($p < 0.05$) positively related to bacterial species richness and diversity estimates such as Ace, Chao1 and Shannon index [46]. The inconsistent effects of soil pH on microbial alpha diversity might be a result along soil pH gradient. There was universal inhibition of all microbial variables below pH 4.5, probably because the release of free aluminum limited microbial growth in acidic soils [47]. In addition, the bacterial alpha diversity in the experimental plots was significantly ($P < 0.05$) higher that of the fungi (Table 4). Inherently much lower fungal diversity might be mainly caused by the growth-inhibiting effects of bacteria on fungi [48]. These findings indicate that different water and fertilizer treatments have a minor influence on microbial alpha diversity in acid paddy soils. However, microbial community structures were greatly affected by water and fertilizer treatments (Fig 5A and 5B), which was consistent with previous observations of a strong influence of N fertilization on microbial community composition [49]. The bacterial phyla *Actinobacteria*, *Cyanobacteria*, *Verrucomicrobia* and fungal phylum *Mucoromycota* were highly favoured, whereas the bacterial phylum *Acidobacteria* was repressed in the T2 treatment (Fig 6A and 6B). Thus, different irrigation and fertilization treatments altered soil microbial community structure, but not their alpha diversity in the paddy soil.

## The influence of environmental factors on nitrogen and phosphorus losses

Different water and fertilizer treatments also altered microbial community structure (Fig 5A and 5B). This agrees with a previous report showing the change in microbial community structure might be caused by their responses to variations in soil properties associated with integrated water and fertilizer management [50]. In this study, the predominant factors controlling soil bacterial community structure were soil pH and Olsen P, while the main factors governing fungal community structure were pH and TN in different water and fertilizer treatments by using db—RDA (Fig 7A and 7B). The alterations in microbial community composition, in turn could affect N and P losses from paddy fields [17]. The ability of *Firmicutes* to fix $N_2$ was used to produce large amounts of $NH_4^+$–N during growth as a well-known potential source of N for rice plants [51,52], which corresponded to the increased N uptake and runoff loss in the T1 and T2 treatments (Tables 2 and 5). In addition, *Bacteroidetes* bacteria *as r—* strategists [53], might be favored by higher soil fertility associated with N and P fertilizer application in the T1 and T2 treatments compared to that in the T0 treatment (Fig 6A). Moreover, *Bacteroidetes* belonged to one of the dominant denitrifiers that had a capacity for the reduction of $NO_3^-$–or $NO_2^-$–N to $N_2$ as the end product in paddy soils, which increased soil TN, especially $NO_3^-$–N loss through surface runoff from paddy fields (Table 5) [54]. *Proteobacteria* was abundant and mainly included free-living N-fixing β-Proteobacteria [55], which provided an efficient N source for paddy soils and thus increased N runoff loss (Table 5). Conversely, *Chloroflexi* belonged to green bacteria, which was a diverse group of chlorophototrophic organisms. Most of these organisms synthesized bacteriochlorophylls c, d or e and utilized chlorosomes for light harvesting, and consequently improved rice growth and productivity [56]. This improved growth characteristics have stimulated root distribution and uptake of N

and P nutrients, which led to a reduction in nutrient runoff losses from paddy fields (Table 5). In addtion, *Planotomycetes*, as oligotrophic bacteria, would be likely stimulated under nutrient-poor conditions, but their growth was inhibited by N and/or P inputs (Fig 6A) [53,57,58]. Moreover, some members of these anammox *planctomycetes* performed ammonium oxidation anaerobically, which led to an increase in $NO_3^-$-N in the T0—treated soil, and thus actually increased $NO_3^-$-N runoff loss risk [59]. An exception was *Nitrospirae* bacteria, which was present at the relatively lower abundance in the T2 treatment than that in the other two treatments (Fig 6A). This result is inconsistent with previous work which has shown that *Nitrospirae* was the dominant bacterial group under combined application of mineral and organic fertilizers in an irrigated farmland [60]. One possible explanation is that the periodic drought of the soils during the entire rice growing season in the T2 treatment, leading to an aerobic environment, especially in the harvest season, may significantly inhibit this facultatively anaerobic chemoautotrophic nitrite oxidizer [61]. Furthermore, *Nitrospirae*, as an ammonia-oxidizing bacterium, had high potential nitrification rates, thereby increasing $NO_3^-$–N runoff loss [18]. Our results further demonstrated that T2 had a small $NO_3^-$–N loss in surface runoff partly because of the low abundance of *Nitrospirae* (Table 5). The *Actinobacteria* were involved in supplying P to plants [62], which corresponded to the increased AEP (agronomic P use efficiency) in the T2 treatment owning to an increase in the abundance of *Actinobacteria*, and thus decreased P loss in surface runoff (Table 5 and Fig 6A). Likewise, some *cyanobacterial* taxa could also drive P cycling by accessing pools of P that are not generally available to plants [63]. The ability of *Cyanobacteria* contributed to increase AEP with an increase in their relative abundance in the T2 treatment, but simultaneously aggravated the P runoff loss (Table 5 and Fig 6A).

The dominant *Ascomycota* fungi has been described as litter decomposers [31], which increased soil N and P contents, in turn, accelerated nutrient runoff losses from paddy fields (Table 5). In addition, *Chytridiomycota* has been reported to infect AMF spores [64]. Moreover, AMF promoted soil aggregate formation, which could protect organic N and P against decomposition from soil microbes [65] and consequently reduced N and P runoff losses (Table 5). *Mucoromycota*, as a saprotroph, most of them could degrade C sources ranging from simple sugars to pectins, hemicelluloses, lipids and proteins when colonizing different substrata [66]. The organic C degradation resulted in higher rice grain yields and TN uptake levels in the T1 and T2 treatments than those in the T0 treatment, and consequently reduced N runoff loss (Tables 2, 3 and 5).

Soil $NH_4^+$–N and $NO_3^-$–N contents were also the dominant impact factors to interpret the difference of N runoff loss among the treatments, followed by N fertilizer input, while the most important factor affecting P runoff loss was P fertilizer input, and secondly, they were soil Olsen P and TP (Fig 9A and 9B). Similarly, it has been reported that soil N pool contributed more than fertilizer input to increased N runoff loss, whereas fertilizer P input contributed more than soil P pool to increased P runoff loss [67]. Hence, these studies further demonstrated that N and P runoff losses were predominantly governed by edaphic factors and fertilization levels during rice-growing season under different water and fertilizer managements. Overall, the integrated strategy for rice irrigation and irrigation might play a major role in shaping soil microbial community structure by altering edaphic properties, which was responsible for N and P losses through surface runoff in paddy soils of subtropical China.

## Conclusions

Our results demonstrated that the T2 (water-saving irrigation and optimizing fertilization) treatment increased agronomic N use efficiency and rice grain yield in the double rice cropping system, which reduced N runoff loss compared to the T1 (traditional irrigation and

fertilization practice) treatment. The N loss in surface runoff was mainly in the form of nitrate N ($NO_3^-$–N) in all treatments. Furthermore, high N fertilizer input, soil $NO_3^-$–N, and ammonium N ($NH_4^+$–N) contents were important contributors to the N loss. In addition, different water and fertilizer treatments caused variations in soil microbial community structure, which might further affect N runoff loss. *Bacteroidetes*, *Proteobacteria*, *Planotomycetes*, *Nitrospirae*, *Firmicutes* bacteria and *Ascomycota* fungi contributed to an increase in the N runoff loss, but the N loss decreased by *Chytridiomycota* fungi. In summary, the T2 treatment should be a cost-effective and environmentally-friendly alternative to traditional fertilization and irrigation method in the present study.

## Supporting information

**S1 Fig.** Bacterial (A) and fungal (B) Shannon–Wiener curves for normalized number of reads at a 97% threshold in different fertilization and irrigation regimes. Notes: T0 = Traditional irrigation; T1 = Traditional irrigation and fertilization practice; T2 = Water-saving irrigation and optimizing fertilization.
(TIF)

## Acknowledgments

The authors would like to thank Professor Juhua Yu (Soil and Fertilizer Institute, Fujian Academy of Agricultural Sciences) for helping to revise languages and conduct experiments.

## Author Contributions

**Conceptualization:** Limin Wang, Dongfeng Huang.

**Data curation:** Limin Wang, Dongfeng Huang.

**Formal analysis:** Limin Wang, Dongfeng Huang.

**Funding acquisition:** Limin Wang, Dongfeng Huang.

**Investigation:** Limin Wang, Dongfeng Huang.

**Methodology:** Limin Wang, Dongfeng Huang.

**Project administration:** Limin Wang, Dongfeng Huang.

**Resources:** Limin Wang, Dongfeng Huang.

**Software:** Limin Wang, Dongfeng Huang.

**Supervision:** Limin Wang, Dongfeng Huang.

**Validation:** Limin Wang, Dongfeng Huang.

**Visualization:** Limin Wang, Dongfeng Huang.

**Writing – original draft:** Limin Wang, Dongfeng Huang.

**Writing – review & editing:** Limin Wang.

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
