## [Decision Letter · Decision Letter 0]

7 Jan 2021

PONE-D-20-38799

Water and fertilizer managements affect nitrogen and phosphorus losses by surface
runoff and microbial communities in a paddy soil

PLOS ONE

Dear Dr. Huang,

Thank you for submitting your manuscript to PLOS ONE. After careful consideration, we
feel that it has merit but does not fully meet PLOS ONE’s publication criteria as it
currently stands. Therefore, we invite you to submit a revised version of the
manuscript that addresses the points raised during the review process.

Please submit your revised manuscript by Feb 18 2021 11:59PM. If you will need more
time than this to complete your revisions, please reply to this message or contact
the journal office at plosone@plos.org. When
you're ready to submit your revision, log on to https://www.editorialmanager.com/pone/ and select the 'Submissions
Needing Revision' folder to locate your manuscript file.

If you would like to make changes to your financial disclosure, please include your
updated statement in your cover letter. Guidelines for resubmitting your figure
files are available below the reviewer comments at the end of this letter.

We look forward to receiving your revised manuscript.

Kind regards,

Dayong Zhao, Ph.D.

Academic Editor

PLOS ONE

Journal Requirements:

2. Please include your tables as part of your main manuscript and remove the
individual files. Please note that supplementary tables should be uploaded as
separate "supporting information" files.

Reviewers' comments:

Reviewer's Responses to Questions

**Comments to the Author**

1. Is the manuscript technically sound, and do the data support the conclusions?

Reviewer #1: Partly

Reviewer #2: Partly

2. Has the statistical analysis been performed
appropriately and rigorously? 

Reviewer #1: Yes

Reviewer #2: Yes

3. Have the authors made all data underlying the
findings in their manuscript fully available?

Reviewer #1: Yes

Reviewer #2: Yes

4. Is the manuscript presented in an intelligible
fashion and written in standard English?

Reviewer #1: Yes

Reviewer #2: No

5. Review Comments to the Author

Reviewer #1: Evaluation of nutrient losses from paddy soils under different
irrigation and fertilization practices is important for sustainable management.
Authors tried to assess nutrient losses under different water and fertilizer
managements and to identify underlying mechanisms. However, the part of results and
discussions are not robust. Besides, no significant highlight is established in the
present form of manuscript. The present form of this manuscript is not recommended
for publication in this journal.

The paper seems technically sound, but I have some doubts regarding the scientific
validity and the rigor in science.

(1) Surface runoff should be primarily determined by water depth in the field and the
size of rainfall events, rather than soil microorganism composition. In general, a
higher water level results in more nutrient losses through surface runoff. Compared
with the water depth (1-6 cm) in T1 (traditional irrigation), the water depth (-3-3
cm) in T2 (shallow intermittent irrigation) was relatively lower. Therefore, the
risk of nutrient loss under T1 should be higher than it under T2. Hence, the major
result of the decline of nutrient losses in T2 is not significant and surprised.

(2) There is a lack of figure, such as SEM, which comprehensively shows the direct
effects of various environmental factors on nutrient losses, and its indirect
effects through altering microorganism community composition. Besides studied
environmental variables (Table 2), the size of rainfall events should be considered
in investigating influencing factors of nutrient losses.

Specific comments:

Abstract

(1) The effect of microorganism community composition on nutrient runoff losses
should be demonstrated specifically.

(2) More details about T1 and T2 should be given.

Introduction

(1) Line 55-57, the relationship between ammonia oxidation and nitrogen losses should
be elaborated.

(2) Line 81, the objective of this study was only to verify an existing irrigation
practice. The “develop” is not appropriate.

Materials and methods

(1) Line 107, why did you choose 20 cm as the height of the concrete cement border? I
think the quantity of runoff is mainly depended on the height of the border.

(2) Line 109, it’s better to show a figure of the runoff collection device.

Results

(1) Line 211, there are too many significant numbers.

(2) The content of 3.5 part is not closely related to the main purpose of this study.
The effects of abiotic factors on the microbial community composition should not be
demonstrated in an independent part. The relationship between microorganisms and
nutrient losses should be addressed specifically.

Discussion

(1) Line 285-286, please keep constant citation format in the text. Wang et a. (2019)
is different from others.

(2) Line 285, the sentence “NO3--N was the main form of TN” doesn’t seem
scientific.

(3) Line 313, please delete one “the”.

(4) About 4.3 part, the relationship between microorganisms and environmental factors
is not the critical issue in this study. Please rewrite and rename this part.

Tables and Figures

(1) Only fertilization treatment was given in the Table 1. Please add irrigation
treatment.

(2) All tables and figures should be improved. The number and letters in the figures
are too small.

(3) About Fig. 2 B, Y-axis number “0.000-8.000” should be changed into “0-8”. Please
correct other similar issues.

(4) Figure resolution need to be improved.

Reviewer #2: Dear Editor and authors,

PONE-D-20-38799 entitled “Water and fertilizer managements affect nitrogen and
phosphorus losses by surface runoff and microbial communities in a paddy soil” have
done nice work but being poorly presented. The MS maybe accepted after major
revision as followings:

1. The language should be revised deeply through the MS, even in Title, please ask a
specialist for help. The title should be “Nitrogen and phosphorus losses by surface
runoff and soil microbial communities in a paddy field with different irrigation and
fertilization managements”

2. Line 14, please clearly define of T1.

3. Line 28 What is the optimum rice yield?

4. Line 36 the total cultivated area in where?

5. In the introduction parts, the authors should clearly figure out the novelty of
your study as there were plenty of literature on N and P losses from paddy fields
with different water and fertilization managements.

6. Line 77 This knowledge gap is what? This should be clearly stated.

7. Table 1’ title should be Fertilization schemes for the different treatments. But
the authors said that “The water and fertilizer practices used in this experiment
are described in Table 1”, where is water managements? On other words, which kind of
water-saving irrigation method you used in this study? Please clearly shown your
experimental design, which is vital for the readers.

8. 2.2 Water sampling methods should be described in details.

9. Results this parts should be clearly and concise. Line 198 (2795 ± 165 t ha−1)
should be deleted as the value has been shown in the Table 2, so many presentation
in the results.

10. Why you only shown the relate data obtained from 2018? If you want to make
concise, please at least to state this in the Data analysis.

11. Do not simply repeat the results in the Discussion parts.

12. The authors mentioned that “Runoff DN loss was mainly in the form of NO3-–N than
in NH4+–N under different water and fertilizer treatments” and given the reasons. I
thinks the differences in N and P losses among the different treatments was your
mainly focus, and should be deeply discussed.

13. Is there any relationship between soil physiochemical properties and N and P
losses? This should be one important novelty of this study.

14. The conclusions should be further clarified.

15. All figures need to be clearly presented, and a lot of texts in figures are not
visible.

Best wishes!

6. PLOS authors have the option to publish the peer
review history of their article (what does this mean?). If published, this will
include your full peer review and any attached files.

If you choose “no”, your identity will remain anonymous but your review may still be
made public.

**Do you want your identity to be public for this peer review?** For
information about this choice, including consent withdrawal, please see our
Privacy Policy.

Reviewer #1: No

Reviewer #2: No

---

## [Author Response · Author response to Decision Letter 0]

17 Mar 2021

Reviewer #1: 

The paper seems technically sound, but I have some doubts regarding the scientific
validity and the rigor in science.

(1) Surface runoff should be primarily determined by water depth in the field and the
size of rainfall events, rather than soil microorganism composition. In general, a
higher water level results in more nutrient losses through surface runoff. Compared
with the water depth (1-6 cm) in T1 (traditional irrigation), the water depth (-3-3
cm) in T2 (shallow intermittent irrigation) was relatively lower. Therefore, the
risk of nutrient loss under T1 should be higher than it under T2. Hence, the major
result of the decline of nutrient losses in T2 is not significant and surprised.

(2) There is a lack of figure, such as SEM, which comprehensively shows the direct
effects of various environmental factors on nutrient losses, and its indirect
effects through altering microorganism community composition. Besides studied
environmental variables (Table 2), the size of rainfall events should be considered
in investigating influencing factors of nutrient losses.

I have added the Fig 9A-B to comprehensively show the direct effects of various
environmental factors on nutrient losses. In addition, runoff volume and fertilizer
input were considered as the influencing factors of nutrient losses in Fig 9A-B.

Abstract

(1)The effect of microorganism community composition on nutrient runoff losses should
be demonstrated specifically.

The effect of microorganism community composition on nutrient runoff losses was
demonstrated specifically (lines 22-25 of revised manuscript with track
changes).

(2) More details about T1 and T2 should be given.

More details about T1 and T2 have been given in lines 16-18, that is T2 (Chemical
fertilizer of 240 kg N ha−1, 120 kg P ha−1, and 240 kg K ha−1 combined with shallow
intermittent irrigation) and T1 (Chemical fertilizer of 273 kg N ha−1, 135 kg P
ha−1, and 135 kg K ha−1 combined with traditional flooding irrigation).

Introduction

(1)Line 55-57, the relationship between ammonia oxidation and nitrogen losses should
be elaborated.

Additionally, ammonia-oxidizing bacteria (AOB) played an important role in the
ammonia oxidation which was crucial for N and P runoff losses [18]. Wang et al.
(2017) found that the ammonia oxidation contributed to 37.5–67.6% of N losses in the
phreatic zone, where AOB might be the major source of nitrite nitrogen (NO2-–N) for
ammonia-oxidizing bacteria [19] (lines 56-59).

(2) Line 81, the objective of this study was only to verify an existing irrigation
practice. The “develop” is not appropriate.

The “develop” was deleted in line 82. 

Materials and methods

(1)Line 107, why did you choose 20 cm as the height of the concrete cement border? I
think the quantity of runoff is mainly depended on the height of the border.

That was because the height of the ridge is 20 cm. 

(2) Line 109, it’s better to show a figure of the runoff collection device.

A figure of the runoff collection device was shown in Fig 1A-B. 

Results

(1)Line 211, there are too many significant numbers.

Many significant numbers were deleted in line 229. 

(2) The content of 3.5 part is not closely related to the main purpose of this study.
The effects of abiotic factors on the microbial community composition should not be
demonstrated in an independent part. The relationship between microorganisms and
nutrient losses should be addressed specifically.

The content of 3.5 part is modified to “Factors impacting N and P surface runoff
losses”, and thus rewrite this part. In addition, the relationship between
microorganisms and nutrient losses should be addressed specifically (lines
294-306).

Discussion

(1)Line 285-286, please keep constant citation format in the text. Wang et a. (2019)
is different from others.

Citation format was modified to constant citation format in the text (Lines 323-324
of revised manuscript with track changes).

(2)Line 285, the sentence “NO3--N was the main form of TN” doesn’t seem
scientific.

The sentence was modified to “NO3−–N was the major form of N in the surface runoff”
in line 325.

(3)Line 313, please delete one “the”.

One “the”was deleted in line 348. 

(4) About 4.3 part, the relationship between microorganisms and environmental factors
is not the critical issue in this study. Please rewrite and rename this part.

The content of 4.3 part was changed into “The influence of environmental factors on
nitrogen and phosphorus losses”, and thus rewrite this part (lines 379-435).

Tables and Figures

(1)Only fertilization treatment was given in the Table 1. Please add irrigation
treatment.

Fertilization and irrigation treatment was given in the Table 1 in lines 114-115.

(2)All tables and figures should be improved. The number and letters in the figures
are too small.

All tables and figures were improved according to your journal requirements.

(3)About Fig. 2 B, Y-axis number “0.000-8.000” should be changed into “0-8”. Please
correct other similar issues.

Too many decimal places In Fig. 2B and other similar issues were corrected.

(4) Figure resolution need to be improved.

Ensure that our images have a resolution of at least 300 pixels per inch (ppi)
according to the full details of the requirements of figure preparation
guidelines.

Reviewer #2: Dear Editor and authors,

PONE-D-20-38799 entitled “Water and fertilizer managements affect nitrogen and
phosphorus losses by surface runoff and microbial communities in a paddy soil” have
done nice work but being poorly presented. The MS maybe accepted after major
revision as followings:

1.The language should be revised deeply through the MS, even in Title, please ask a
specialist for help. The title should be “Nitrogen and phosphorus losses by surface
runoff and soil microbial communities in a paddy field with different irrigation and
fertilization managements”

The title and the language in the text were revised deeply through the MS withe track
changes. 

2.Line 14, please clearly define of T1.

More details about T1 and T2 have been given in lines 16-18, that is T2 (Chemical
fertilizer of 240 kg N ha−1, 120 kg P ha−1, and 240 kg K ha−1 combined with shallow
intermittent irrigation) and T1 (Chemical fertilizer of 273 kg N ha−1, 135 kg P
ha−1, and 135 kg K ha−1 combined with traditional flooding irrigation).

3.Line 28 What is the optimum rice yield?

The “optimum” was deleted in line 29.

4.Line 36 the total cultivated area in where?

Rice (Oryza sativa L.) is one of the main staple crops and feeds over 65% of the
world’s population with 11% of cultivated land [1-2] (lines 32-33). 

5.In the introduction parts, the authors should clearly figure out the novelty of
your study as there were plenty of literature on N and P losses from paddy fields
with different water and fertilization managements.

To date, N and P runoff losses and their influencing factors while maintaining or
enhancing rice yields in the paddy fields in southeastern China are currently
unclear under different irrigation and fertilization regimes. Thus, we hypothesized
that the appropriate irrigation and fertilization practices could affect N and P
runoff losses by environmental factor variations. 

6.Line 77 This knowledge gap is what? This should be clearly stated.

 N and P runoff losses and their influencing factors while maintaining or enhancing
rice yields in the paddy fields in southeastern China are currently unclear under
different irrigation and fertilization regimes. 

7. Table 1’ title should be Fertilization schemes for the different treatments. But
the authors said that “The water and fertilizer practices used in this experiment
are described in Table 1”, where is water managements? On other words, which kind of
water-saving irrigation method you used in this study? Please clearly shown your
experimental design, which is vital for the readers.

Fertilization and irrigation treatment was given in the Table 1 in lines 114-115.

8. 2.2 Water sampling methods should be described in details.

The detail description of water sampling methods is in lines 119-122.

9. Results this parts should be clearly and concise. Line 198 (2795 ± 165 t ha−1)
should be deleted as the value has been shown in the Table 2, so many presentation
in the results.

Many presentation, such as (2795 ± 165 t ha−1) in the results have been deleted.

10. Why you only shown the relate data obtained from 2018? If you want to make
concise, please at least to state this in the Data analysis.

We wanted to make concise, and stated this in the Data analysis line 189.

11. Do not simply repeat the results in the Discussion parts.

The simply repeated results in the Discussion parts have been modified or deleted
with track changes in the text.

12. The authors mentioned that “Runoff DN loss was mainly in the form of NO3-–N than
in NH4+–N under different water and fertilizer treatments” and given the reasons. I
thinks the differences in N and P losses among the different treatments was your
mainly focus, and should be deeply discussed.

The reasons were given (lines 325-328). Moreover, the differences in N and P losses
among the different treatments was deeply discussed (lines 348-356).

13.Is there any relationship between soil physiochemical properties and N and P
losses? This should be one important novelty of this study.

The redundancy analysis (RDA) was conducted to determine which soil variables were
related to N and P losses in Fig 9A-B, and described specially in lines 303-310 and
427-435. 

14. The conclusions should be further clarified.

The conclusions were further clarified in lines 438-448.

15. All figures need to be clearly presented, and a lot of texts in figures are not
visible.

Ensure that our images have a resolution of at least 300 pixels per inch (ppi)
according to the full details of the requirements of figure preparation
guidelines.

to Reviewers.doc
---

## [Decision Letter · Decision Letter 1]

19 May 2021

PONE-D-20-38799R1

Nitrogen and phosphorus losses  by surface runoff and soil microbial communities in a
paddy field with different irrigation and fertilization managements

PLOS ONE

Dear Dr. Huang,

Thank you for submitting your manuscript to PLOS ONE. After careful consideration, we
feel that it has merit but does not fully meet PLOS ONE’s publication criteria as it
currently stands. Therefore, we invite you to submit a revised version of the
manuscript that addresses the points raised during the review process.

I appreciate the revisions made. Although two reviewers agreed to accept the article,
some additional comments are provided from an editorial standpoint. Most are rather
specific and should be easy to address.

Please submit your revised manuscript by Jul 03 2021 11:59PM. If you will need more
time than this to complete your revisions, please reply to this message or contact
the journal office at plosone@plos.org. When
you're ready to submit your revision, log on to https://www.editorialmanager.com/pone/ and select the 'Submissions
Needing Revision' folder to locate your manuscript file.

If you would like to make changes to your financial disclosure, please include your
updated statement in your cover letter. Guidelines for resubmitting your figure
files are available below the reviewer comments at the end of this letter.

We look forward to receiving your revised manuscript.

Kind regards,

Dayong Zhao, Ph.D.

Academic Editor

PLOS ONE

Journal Requirements:

Additional Editor Comments (if provided):

I appreciate the revisions made. Although two reviewers agreed to accept the article,
some additional comments are provided from an editorial standpoint. Most are rather
specific and should be easy to address.

Specific comments:

Line 14. Change “Therefore” to “Here or In this study”.

Line 34. Please provide the full name of the abbreviation of N and P, though they
have been defined in the abstract section.

Line 36-37. Please cite one reference for this sentence.

Line 37. Please add “,” between “To date” and “water-quality”.

Line 49. Change “For example, Chen et al. (2018) reported that” to “It has been
reported that”.

Line 53. Change “For instance,” to “Related studies have suggested that”.

Lien 57-60. Please consider the deletion of these sentences.

Line 64-66. We all know that high-throughput sequencing is widely used in determining
the diversity and composition of soil microbes. These sentences did not provide very
useful information. Thus, please consider removing the sentences.

Line 67. Remove the “Moreover”.

Line 79-84. According to the title and abstract, different irrigation and
fertilization practices could affect soil physicochemical properties and
correspondingly influence soil microbial communities, and thereby contribute to N
and P runoff losses. Am I right? However, this part has nothing to do with soil
microorganisms. Your hypothesis also does not relate to microorganisms.

Line 174-183. What did you do with singleton sequences (i.e., sequence appearing only
one time in the entire data set)? Moreover, how do you address uneven sequencing
depth across samples? Please be clearer in your presentation.

Line 176-181. Please provide the versions of UPARSE, Silva rRNA database, Mothur
software and insert references for them.

Line 179. The authors mentioned that you obtained rarefaction curves using Mothur
software, where is the result of rarefaction curves?

Line 193 and Line 200. Both Rstudio and R were used in this study to perform
statistical analyses. Please provide their versions in your manuscript and insert
reference.

Line 194. Provide the version of the R pheatmap package and cite one reference.

Line 199-200. Which distance did you use? Please make it clear.

Line 197. The “vif” is an abbreviation form. Please define it at its first
mention.

Line 200. Provide the version of R vegan package, and then insert one reference for
it.

Line 241. Double “the”.

Line 263-267. According to the description of microbial alpha diversity indices in
Line 180, change “Ace” to “ACE” and change “Chao” to “Chao1” in the table 4.
Meanwhile, provide the explanation of ACE in the table notes in Line 264-267.

Line 306-307. Other environmental factor refer to what? And they would directly
affect the N and P losses. I think this sentence can be removed as what is needed in
the results section is for the author to describe their findings objectively.

Figures

Figure 1. This image needs to be cropped appropriately as there is a lot of white
space throughout the image.

Figure 3. As an example, the font of the words “Runoff yield” and “TN loss loads” in
the vertical titles did not seem to correspond to the font of the horizontal title.
Please unify the font of all the text in the figure A, B, C and D. In addition,
there are up ticks and right ticks in the X-axis and Y-axis of Fig. 3D,
respectively, while all ticks in Fig. 3A, 3B and 3C are not shown. Please unify the
drawing style. As for Fig. 3D, my suggestion is that the authors can show down ticks
and left ticks in X-axis and Y-axis, respectively.

Figure 3B. A point to note in the illustration of the types of line in the image is
that the presentation of "NO3--N" and "NH4+-N" should be revised as they are not
presented in a very aesthetically pleasing way.

Figure 3D. There are no data points between 0 and 0.6 mg/L of total phosphorus
concentration, so the authors could have left the values of the horizontal
coordinates not starting from 0.

Figure 4. The word “Venn” and three solid dots with “T0/T1/T2” could be removed from
your figure as they are redundant. In addition, the size of the words (e.g., T0, T1
and T2) and numbers in the Venn plots needs to be slightly adjusted upwards. Figure
4A: The bar diagram has two ticks at the value 200 of the vertical coordinate,
please deal with them.

Figure 5. Please delete the word “PLS-DA on OTU level” at the top of the figure.

Figure 6. Only major phyla are presented in each treatment. What is the relative
abundance of phyla below which they are classified as 'Others'? This needs to be
made clear in the figure caption. In addition, please delete the word “Community
barplot analysis” at the top of the figure.

Figure 7. Please delete the word “db-RDA on OTU level” at the top of the figure.

Figure 8. Please delete the word “Spearman Correlation Heatmap” at the top of the
figure.

Figure 3-Figure 9: The numbers 0, 1 and 2 are below the letter T (i.e., the form of a
subscript) in the figure captions and main text. However, in the figure, the author
does not show the T0, T1 and T2 in a subscript form, so please standardize the
format of presentation.

Reviewers' comments:

Reviewer's Responses to Questions

**Comments to the Author**

1. If the authors have adequately addressed your comments raised in a previous round
of review and you feel that this manuscript is now acceptable for publication, you
may indicate that here to bypass the “Comments to the Author” section, enter your
conflict of interest statement in the “Confidential to Editor” section, and submit
your "Accept" recommendation.

Reviewer #1: All comments have been addressed

Reviewer #2: (No Response)

2. Is the manuscript technically sound, and do the data
support the conclusions?

Reviewer #1: Yes

Reviewer #2: Yes

3. Has the statistical analysis been performed
appropriately and rigorously? 

Reviewer #1: Yes

Reviewer #2: Yes

4. Have the authors made all data underlying the
findings in their manuscript fully available?

Reviewer #1: Yes

Reviewer #2: Yes

5. Is the manuscript presented in an intelligible
fashion and written in standard English?

Reviewer #1: Yes

Reviewer #2: Yes

6. Review Comments to the Author

Reviewer #1: (No Response)

Reviewer #2: (No Response)

7. PLOS authors have the option to publish the peer
review history of their article (what does this mean?). If published, this will
include your full peer review and any attached files.

If you choose “no”, your identity will remain anonymous but your review may still be
made public.

**Do you want your identity to be public for this peer review?** For
information about this choice, including consent withdrawal, please see our
Privacy Policy.

Reviewer #1: No

Reviewer #2: No

---

## [Author Response · Author response to Decision Letter 1]

16 Jun 2021

Additional Editor Comments (if provided):

I appreciate the revisions made. Although two reviewers agreed to accept the article,
some additional comments are provided from an editorial standpoint. Most are rather
specific and should be easy to address.

Specific comments:

Line 14. Change “Therefore” to “Here or In this study”.

 “Therefore”was changed into “Here”in line 14 of revised manuscript with track
changes.

Line 34. Please provide the full name of the abbreviation of N and P, though they
have been defined in the abstract section.

We provided the full name of the abbreviation of N and P in line 33 of revised
manuscript with track changes.

Line 36-37. Please cite one reference for this sentence.

We cited one reference for this sentence in line 37.

Line 37. Please add “,” between “To date” and “water-quality”.

We added “,” between “To date” and “water-quality”in line 37.

Line 49. Change “For example, Chen et al. (2018) reported that” to “It has been
reported that”.

We Changed “For example, Chen et al. (2018) reported that” to “It has been reported
that” in line 48.

Line 53. Change “For instance,” to “Related studies have suggested that”.

We Changed “For instance,” to “Related studies have suggested that”in line 51.

Line 57-60. Please consider the deletion of these sentences.

We deleted these sentences of lines 57-60. 

Line 64-66. We all know that high-throughput sequencing is widely used in determining
the diversity and composition of soil microbes. These sentences did not provide very
useful information. Thus, please consider removing the sentences.

We removed the sentences of lines 64-66. 

Line 67. Remove the “Moreover”.

We removed the “Moreover”in line 67. 

Line 79-84. According to the title and abstract, different irrigation and
fertilization practices could affect soil physicochemical properties and
correspondingly influence soil microbial communities, and thereby contribute to N
and P runoff losses. Am I right? However, this part has nothing to do with soil
microorganisms. Your hypothesis also does not relate to microorganisms.

We Changed this part to “Thus, we hypothesized that different irrigation and
fertilization practices could alter soil chemical properties and microbial community
structure, which would subsequently affect N and P runoff losses. To test the
hypothesis, a 10-year plot experiment was conducted to estimate N and P runoff
losses and uptake, soil chemical properties, microbial diversity, and community
composition under different fertilization and irrigation regimes. In general, the
purpose of this study was to ⑴ verify an optimal irrigation and fertilization
practice in order to minimize N and P runoff losses, and ⑵ explore the factors
influencing N and P losses in surface runoff from paddy fields in southeastern
China.” in lines 71-77.

Line 174-183. What did you do with singleton sequences (i.e., sequence appearing only
one time in the entire data set)? Moreover, how do you address uneven sequencing
depth across samples? Please be clearer in your presentation.

We revised this part according to editors' requirements in lines 169-181.
Specifically, Using the UPARSE (version 7.0.1090), we also removed singleton
sequences (i.e., sequences appearing only one time in the entire data set). As the
number of sequence reads in each sample varied, the OTU table was rarified (holding
the same sequence number in each sample) prior to microbial community diversity
calculations.

Line 176-181. Please provide the versions of UPARSE, Silva rRNA database, Mothur
software and insert references for them.

We provided the versions of UPARSE, Silva rRNA database, Mothur software and insert
references for them in lines 171-179.

Line 179. The authors mentioned that you obtained rarefaction curves using Mothur
software, where is the result of rarefaction curves?

We added the result of rarefaction curves in lines 251-253.

Line 193 and Line 200. Both Rstudio and R were used in this study to perform
statistical analyses. Please provide their versions in your manuscript and insert
reference.

We provided the versions of R in line 191 and line 199.

Line 194. Provide the version of the R pheatmap package and cite one reference.

We provided the versions of R in line 199.

Line 199-200. Which distance did you use? Please make it clear.

The unweighted UniFrac distance - based redundancy analysis (db‐RDA) was processed by
R software (version 2.15.3) in lines 198-199.

Line 197. The “vif” is an abbreviation form. Please define it at its first
mention.

We defined the “vif” as variance inflation factor in line 195.

Line 200. Provide the version of R vegan package, and then insert one reference for
it.

We provided the versions of R in line 199, and then inserted one reference for
it.

Line 241. Double “the”.

We deleted “the”in line 240.

Line 263-267. According to the description of microbial alpha diversity indices in
Line 180, change “Ace” to “ACE” and change “Chao” to “Chao1” in the table 4.
Meanwhile, provide the explanation of ACE in the table notes in Line 264-267.

We changed “Ace” to “ACE” and changed “Chao” to “Chao1” in the table 4. Meanwhile,
provide the explanation of ACE (abundance-based coverage estimators) in the table
notes in Line 265.

Line 306-307. Other environmental factor refer to what? And they would directly
affect the N and P losses. I think this sentence can be removed as what is needed in
the results section is for the author to describe their findings objectively.

This sentence “Other environmental factor refer to what? And they would directly
affect the N and P losses” was removed in this manuscript.

Figures

Figure 1. This image needs to be cropped appropriately as there is a lot of white
space throughout the image.

Figure 1 was cropped appropriately.

Figure 3. As an example, the font of the words “Runoff yield” and “TN loss loads” in
the vertical titles did not seem to correspond to the font of the horizontal title.
Please unify the font of all the text in the figure A, B, C and D. In addition,
there are up ticks and right ticks in the X-axis and Y-axis of Fig. 3D,
respectively, while all ticks in Fig. 3A, 3B and 3C are not shown. Please unify the
drawing style. As for Fig. 3D, my suggestion is that the authors can show down ticks
and left ticks in X-axis and Y-axis, respectively.

Figure 3 was revised according to editors' requirements.

Figure 3B. A point to note in the illustration of the types of line in the image is
that the presentation of "NO3--N" and "NH4+-N" should be revised as they are not
presented in a very aesthetically pleasing way.

"NO3--N" and "NH4+-N" in Figure 3 were revised. 

Figure 3D. There are no data points between 0 and 0.6 mg/L of total phosphorus
concentration, so the authors could have left the values of the horizontal
coordinates not starting from 0.

Figure 3D was revised.

Figure 4. The word “Venn” and three solid dots with “T0/T1/T2” could be removed from
your figure as they are redundant. In addition, the size of the words (e.g., T0, T1
and T2) and numbers in the Venn plots needs to be slightly adjusted upwards. Figure
4A: The bar diagram has two ticks at the value 200 of the vertical coordinate,
please deal with them.

Figure 4 was improved according to editors' requirements.

Figure 5. Please delete the word “PLS-DA on OTU level” at the top of the figure.

The word “PLS-DA on OTU level” at the top of the figure 5 was deleted.

Figure 6. Only major phyla are presented in each treatment. What is the relative
abundance of phyla below which they are classified as 'Others'? This needs to be
made clear in the figure caption. In addition, please delete the word “Community
barplot analysis” at the top of the figure.

Fig 6. Average relative abundance of dominant bacterial (A) and fungal (B) phyla
(> 1.0%) in different fertilization and irrigation regimes. The abundance is
expressed as the average percentage of the targeted sequences to the total
high-quality bacterial and fungal sequences of samples from triplicate plots of each
fertilization regime, respectively. Notes: ‘Others’ refer to those identified phyla
with lower than 1.0% relative abundance in all the samples. T0 = Traditional
irrigation; T1 = Traditional irrigation and fertilization practice; T2 =
Water-saving irrigation and optimizing fertilization. In addition, we deleted the
word “Community barplot analysis” at the top of the figure.

Figure 7. Please delete the word “db-RDA on OTU level” at the top of the figure.

 We deleted the word “db-RDA on OTU level” at the top of the figure.

Figure 8. Please delete the word “Spearman Correlation Heatmap” at the top of the
figure.

We deleted the word “Spearman Correlation Heatmap” at the top of the figure.

Figure 3-Figure 9: The numbers 0, 1 and 2 are below the letter T (i.e., the form of a
subscript) in the figure captions and main text. However, in the figure, the author
does not show the T0, T1 and T2 in a subscript form, so please standardize the
format of presentation.

We standardized the format of presentation of T0, T1 and T2 in both Figure 3-Figure 9
and in this manuscript.

to Reviewers.doc
---

## [Editor Report · Decision Letter 2]

23 Jun 2021

Nitrogen and phosphorus losses by surface runoff and soil microbial communities in a
paddy field with different irrigation and fertilization managements

PONE-D-20-38799R2

Dear Dr. Huang,

We’re pleased to inform you that your manuscript has been judged scientifically
suitable for publication and will be formally accepted for publication once it meets
all outstanding technical requirements.

Kind regards,

Dayong Zhao, Ph.D.

Academic Editor

PLOS ONE
---

## [Editor Report · Acceptance letter]

1 Jul 2021

PONE-D-20-38799R2 

Nitrogen and phosphorus losses by surface runoff and soil microbial communities in a
paddy field with different irrigation and fertilization managements 

Dear Dr. Huang:

I'm pleased to inform you that your manuscript has been deemed suitable for
publication in PLOS ONE. Congratulations! Your manuscript is now with our production
department. 

Kind regards, 

on behalf of

Dr. Dayong Zhao 

Academic Editor

PLOS ONE